# AUDIO TURING TEST: BENCHMARKING THE HUMAN-LIKENESS OF LARGE LANGUAGE MODEL-BASED TEXT-TO-SPEECH SYSTEMS IN CHINESE

## ABSTRACT

Recent advances in large language models (LLMs) have significantly improved text-to-speech (TTS) systems, enhancing control over speech style, naturalness, and emotional expression, which brings TTS Systems closer to human-level performance. Yet evaluation still relies largely on the Mean Opinion Score (MOS), whose subjectivity, environmental variability, and limited interpretability prevent it from faithfully capturing how human-like the synthesized audio is. Existing evaluation datasets also lack a multi-dimensional design, often neglecting factors such as speaking styles, context diversity, and trap utterances, which is particularly evident in Chinese TTS evaluation. To address these challenges, we introduce the **A**udio **T**uring **T**est (ATT), a multi-dimensional Chinese corpus dataset ATT-Corpus paired with a simple, Turing-Test-inspired evaluation protocol. Instead of relying on complex MOS scales or direct model comparisons, ATT asks evaluators to judge whether a voice sounds human. This simplification reduces rating bias and improves evaluation robustness. To further support rapid model development, we also finetune Qwen2.5-Omni-7B with human judgment data as Auto-ATT for automatic evaluation. Experimental results show that ATT effectively differentiates models across specific capability dimensions using its multi-dimensional design. Auto-ATT also demonstrates strong alignment with human evaluations, confirming its value as a fast and reliable assessment tool.

## 1 INTRODUCTION

Achieving human-likeness in speech is now a central objective for modern Text-to-Speech (TTS) systems since the widespread need for human-likeness in applications raises the bar for natural, expressive, and contextually appropriate output (Jain et al., 2025; Wang et al., 2024; Yang et al., 2024b; Yeh et al., 2024). Recent LLM-driven advances have accelerated this pursuit: LLM architectures enrich controllability over style and intonation (Anastassiou et al., 2024; Li et al., 2024) and substantially improve speech naturalness and emotional expressivity (Wang et al., 2025), pushing systems from near-human toward truly human-rivaling performance. To further elevate human-likeness, accurate evaluation is indispensable. As realism improves, the perceptual gaps among state-of-the-art LLM-based TTS systems narrow, making it increasingly difficult to distinguish their performance with coarse metrics or underspecified protocols (Le Maguer et al., 2024). This intensifies the need for reliable, sensitive, and well-calibrated evaluation frameworks that can measure human-likeness, diagnose residual deficiencies, and guide continued model development.

Current TTS evaluation still lacks methods and datasets specifically designed for human-likeness evaluation. Listener-based 5-point Mean Opinion Score (MOS) (International Telecommunication Union, 2018) and variants such as CMOS are broad, aggregate judgments for TTS quality evaluation. These MOS-based methods collapse multiple perceptual dimensions into a single scalar and thus offers limited diagnostic value.

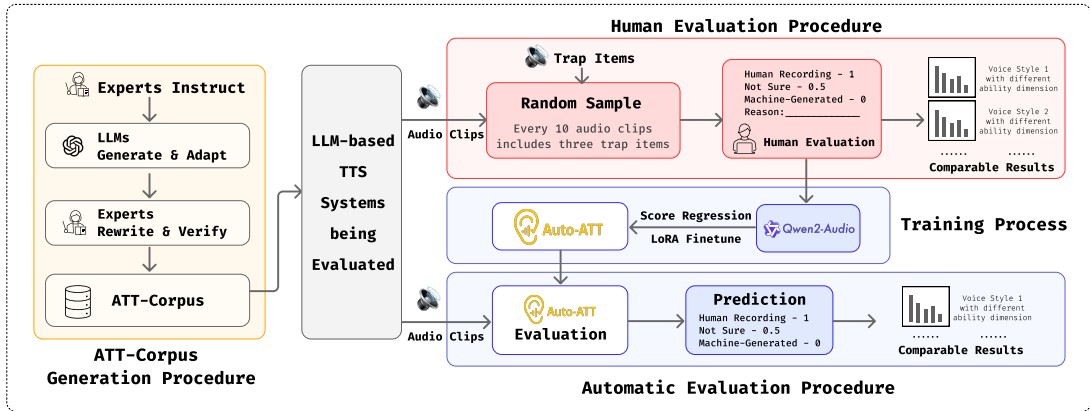

Figure 1: **Audio Turing Test Evaluation Framework**: (1) Corpus Generation: a semi-automatic corpus generation pipeline for generating the challenge TTS synthesis corpus for ATT evaluation; (2) Human Evaluation: a human-evaluation protocol that enables precise, comparable assessments and lowers evaluation costs through a simple yet effective Turing-test-style design, (3) Automatic Evaluation: Auto-ATT, an automatic tool to predict the Human-likeness Score for rapid iterations.

In practice, this makes MOS ill-suited for pinpointing concrete defects and not suitable for assessing the nuanced question of human-likeness. Beyond MOS's known limits, most TTS evaluation corpora remain general-purpose rather than purpose-built to probe multidimensional capabilities (Anastassiou et al., 2024; Wang et al., 2025). Listening tests seldom include hidden human references or crafted trap utterances to diagnose rater bias and attention allocation (Chiang et al., 2023). These gaps are acute for Chinese, where prosodic pauses, multilingual code-switching, polyphonic characters, and special symbols strongly shape fluency and naturalness (Lavin, 2002; Yang et al., 2024a; Dai et al., 2025). Consequently, the lack of multidimensional and trap data in existing datasets compounds MOS-based weaknesses and limits the discriminative power and completeness of current TTS evaluations (Chiang et al., 2023).

Inspired by the classic Turing Test (French, 2000), as shown in Figure 1, we propose the **A**udio **T**uring **T**est (ATT), an evaluation framework combining a multi-dimensional dataset ATT-Corpus with a Turing test-based evaluation protocol and metrics. To evaluate the human-likeness of Chinese TTS systems, we first built a targeted evaluation corpus addressing key challenges in Chinese speech synthesis. Based on the ATT-Corpus, we design a simple and easy-implement human evaluation protocol. By requiring evaluators to provide ternary judgments on whether each sample is human, along with brief justifications, ATT facilitates both quantitative and qualitative assessments of speech human-likeness. This approach mitigates the anchoring effects and lack of cross-context comparability commonly associated with traditional scale-based methods such as MOS. ATT employs randomized clip assignment, trap items for attention monitoring, and expert-validated justifications to ensure data quality, supporting reliable, unbiased clip-level analysis. To enable swift automated evaluation and accelerate TTS model iteration, we fine-tuned Qwen2.5-Omni-7B (Xu et al., 2025) on a rigorously annotated ATT dataset, producing Auto-ATT.

Using the ATT protocol, we collected ratings from 857 native Chinese listeners through crowdsourcing platforms. Experimental results demonstrate that ATT is a sharp and reliable evaluation framework. Benchmarking results further indicate that ATT effectively distinguishes the performance of different TTS models. Notably, even the top-performing model, Seed-TTS (Anastassiou et al., 2024), achieves only a human-likeness score of 0.4 on ATT—considerably lower than that of real human speech, and in stark contrast to previously reported MOS scores. Analyses across sub-dimensions and voice styles demonstrate that ATT enables multi-axis evaluation of LLM-based TTS systems and supports direct cross-system comparisons. We assess the effectiveness of Auto-ATT through trap item tests and by comparing auto-evaluation results with human

ratings. Auto-ATT significantly outperforms traditional MOS predictors in evaluating trap clips and shows strong alignment with human scores.

In summary, our contributions are as follows:

- We introduce the Audio Turing Test, an evaluation framework comprising a multi-dimensional Chinese corpus (ATT-Corpus) and a Turing Test-inspired protocol, designed to effectively assess the human-likeness of LLM-based TTS systems.
- We further train Auto-ATT on human evaluation data to develop an automatic evaluation tool that enables fast and effective assessment of TTS systems, demonstrating its effectiveness through strong consistency with human ratings.
- We benchmark state-of-the-art LLM-based TTS systems using both quantitative and qualitative analyses, thereby validating the effectiveness and robustness of the ATT framework in Chinese human-likeness evaluation.

## 2 RELATED WORKS

The quality of TTS systems is typically assessed with a mix of objective metrics and subjective listening tests. Among objective metrics, speaker similarity (SIM) is widely used in recent LLM-based TTS work (Wang et al., 2023; Anastassiou et al., 2024), but it requires reference speech, limits cross-system benchmarking to model trainers, and only reflects voice matching rather than broader quality attributes (Guner et al., 2012). Learned predictors trained on human labels (e.g., UTMOS, DNSMOS) can estimate perceived quality but often struggle to generalize to new systems (Saeki et al., 2022; Reddy et al., 2022).

Subjective evaluation still relies on Mean Opinion Score (MOS) as the de facto "gold standard," with derivatives such as CMOS, CCR, and MUSHRA-style tests (Streijl et al., 2016; Naderi et al., 2021; International Telecommunication Union, 2015). However, MOS collapses multiple perceptual dimensions (naturalness, intelligibility, prosody, speaker similarity, robustness) into a single coarse rating, hindering diagnostic insight. Empirical studies highlight cross-study incomparability due to inconsistent scales/instructions (Kirkland et al., 2023) and sensitivity to listeners' task assumptions (Edlund et al., 2024; Nguyen & Le, 2025). Comparative protocols are also vulnerable: lower-quality systems can depress or inflate scores of better systems (Le Maguer et al., 2024), MUSHRA's human reference can bias judgments (Varadhan et al., 2024), and CMOS may show weak discrimination when items are similarly rated overall. Pairwise and grouping analyses have shown improved sensitivity for naturalness comparisons (Perrotin et al., 2023).

In practice, reporting of human tests is often under-specified (e.g., screening, compensation, interface instructions), which undermines reproducibility (Chiang et al., 2023). As LLM-era TTS approaches human quality, MOS-based evaluations face ceiling effects (Le Maguer et al., 2024) and insufficient resolution for human-likeness. Since the community has begun to focus on human-centered TTS evaluation (Srinivasa Varadhan et al., 2025), there is thus a pressing need for a human-likeness–oriented methodology—with a clear protocol and multidimensional test sets—to enable precise, reliable, and replicable assessment of TTS systems.

## 3 AUDIO TURING TEST

To address the challenges in the current subjective evaluation of TTS systems, we design the Audio Turing Test (ATT). ATT is an evaluation framework with a standardized human evaluation protocol and an accompanying dataset ATT-Corpus, aiming to resolve the lack of unified protocols in TTS evaluation and the difficulty in comparing multiple TTS systems. Moreover, for comprehensive evaluation, ATT-Corpus is designed with appropriate dimensions to help identify specific capability differences among TTS systems. To further support

Table 1: **Corpus Examples of ATT-Corpus.**

| Dimension | Description | Example |
|---|---|---|
| Special Characters and Numerals | Analyze the numbers, special characters, letters, and other information types in the text and transcribe them into the most appropriate or commonly used pronunciations. | 我们公司也有些年头了呢。2010 年 6 月 8 日的时候公司刚成立，现在算算已经快满 12 年了，真的是时间过得挺快的。这一路走来也不容易啊。 |
| Chinese-English Code-switching | Primarily Chinese, interspersed with a few words from other languages, used to assess whether the pronunciation is accurate. | 没想到B站有这么多不同类型的片子，昨晚我在 bilibili 上看了一部新的纪录片...... |
| Paralinguistic Features and Emotions | Expressive paralinguistic phenomena, such as laughter, and the expression of various emotional states. | 呜呼，终于下班了。今天的工作简直让人崩溃，真是忙得一刻都没停过。溜了溜了，赶紧回家休息了，我感觉一回家就要睡着，等会晚点去个洗脚城好好放松一下。 |
| Classical Chinese Poetry/Prose | Each character in classical Chinese poetry and prose is pronounced correctly in terms of its initial consonant, final, tone, and other aspects of articulation. | 苏辙笔下长江的描绘："出西陵，始得平地，其流奔放肆大。"江水奔腾不息、气势磅礴的景象让人震撼不已。三峡之行...... |
| Polyphonic Characters | Polyphonic Chinese characters are pronounced correctly. | 老中医说，这病症得慢慢调理，着急不得。可这病的症结到底在哪呢？ |

the training and iteration of TTS systems, we utilized additional private evaluation data to train Auto-ATT based on Qwen2.5-Omni-7B via LoRA (Hu et al., 2022) finetuning, enabling a model-as-a-judge approach for rapid evaluation of TTS systems on the ATT-Corpus. In this section, we provide a detailed description of the construction of the ATT-Corpus, ATT evaluation protocol design along with the Auto-ATT.

## 3.1 ATT-CORPUS DATASET

Currently, TTS evaluation primarily relies on a subset of samples selected from publicly available speech datasets. This results in limited coverage and makes assessing a model's ability to synthesize complex speech challenging. We construct ATT-Corpus as a comprehensive corpus for TTS evaluation to address this limitation. Taking Chinese as a representative example, we first identify the key challenges TTS systems face, which guide the two-stage data production process of ATT-Corpus.

**Data Description.** We categorize the linguistic capabilities required for Chinese TTS synthesis based on the linguistic phenomena in the corpus to construct a dataset tailored for ATT evaluation. The corpus covers five key dimensions of Chinese linguistic competence: (1) Special Characters and Numerals, (2) Chinese-English Code-switching, (3) Paralinguistic Features and Emotions, (4) Classical Chinese Poetry/Prose, and (5) Polyphonic Characters. The detailed composition of the corpus is presented in Table 1.

**Corpus Generation and Verification.** To reduce manual labor costs and ensure the long-term sustainability of the corpus production process, we adopt a semi-automated approach that combines initial generation and adaptation using large language models (LLMs), followed by expert revision and validation [1]. We employ GPT-4o (Hurst et al., 2024) as the primary model for initial corpus generation. We generate base corpora across various linguistic categories using the prompt and sample text illustrated in the figure. Subsequently, we utilize DeepSeek-R1 (Guo et al., 2025) to perform colloquial adaptation in Chinese, enhancing the naturalness and human-likeness of the generated text. After the automated generation process, four linguistics experts conducted standardized revisions of the corpus. The prompts for data generation, along with the specific revision and review guidelines, are provided in Appendix A.1. Upon completion of the revisions, the experts conducted cross-checking to ensure the quality of the corpus.

---

[1]Experts refer to individuals holding a master's degree in linguistics or a related field.

## 3.2 EVALUATED AUDIO CLIPS GENERATION AND VALIDATION

After completing the corpus collection, we generate audio clips using the TTS models to be evaluated. To ensure evaluation accuracy, we perform manual spot checks on the synthesized speech with the involvement of two expert reviewers. This validation stage is primarily intended to confirm that no widespread synthesis failures occur due to engineering issues or other extraneous factors. Occasional synthesis failures at the level of a single audio clip are recorded but are not discarded at this stage. To balance the sample's representativeness with the efficient use of human review resources, a sampling rate of 25% is adopted. Specifically, we examine two aspects during this stage: synthesis success and synthesis consistency. The details of the validation process are in Appendix A.2. Note that at this stage, we do not evaluate or inspect the human-likeness of the synthesized speech.

## 3.3 HUMAN EVALUATION PROTOCOL

In the ATT human evaluation, participants completed a forced-choice speech-authenticity test. As shown in Figure 1, we propose the following protocol to implement ATT:

**Sampling and Assignment.** Each participant is randomly assigned seven audio clips sampled without replacement from a pool containing the synthesized audio clips for evaluation.

**Attention Monitoring via Trap Items.** To ensure participant attentiveness, we include trap items at regular intervals. Specifically, three random trap items are assigned to each participant in addition to the seven assigned audio clips for evaluation: one deliberately flawed synthetic clip and two genuine human recordings. We also open source these trap items in ATT-Corpus for future evaluation.

**Labeling and Justification.** For each audio clip, participants select one of three labels: [Human], [Unclear], or [Machine]. They are also required to provide a short free-text justification to support qualitative analysis.

**Attention Check Validation.** The response batch of participants is considered valid only if they correctly identify the deliberately flawed synthetic clip and at least one of the two human recordings within each 10-clip set. Responses that fail to meet this criterion are excluded from further analysis.

**Expert Consistency Review.** After data collection, the two expert reviewers assess whether participants' free-text justifications align with their labels. Experts specifically inspect participants' justifications for the seven non-trap synthetic clips, requiring evidence-based and targeted analysis. Responses flagged as inconsistent by either expert are also excluded.

Each audio clip and its corresponding judgment were treated as an independent sampling unit in our protocol design. The random assignment of audio clips without the in-group comparison, minimized learning effects and reduced inter-trial dependence, enabling clip-level modeling of classification accuracy.

To validate the protocol's effectiveness, we report results from a mixed-effects logistic regression analysis, with participants modeled as a random effect, using a generalized linear mixed model (GLMM) (Bolker et al., 2009).

## 3.4 HUMAN-LIKENESS SCORE

Based on the evaluation protocol, we define a metric to quantify the human-likeness of audio clips synthesized by TTS systems: the Human-likeness Score (HLS).

The HLS relies on one human label for each audio clip $i$ collected in the set $\mathcal{L} = \{\text{Human}, \text{Unclear}, \text{Machine}\}$. In HLS, the individual scores for each audio clip $i$ are then expressed using the indicator function $\mathbb{1}(\cdot)$:

$$s_i = \mathbb{1}(\text{Label} = \text{Human}) + 0.5 \cdot \mathbb{1}(\text{Label} = \text{Unclear})$$

Given $N$ audio clips produced by one TTS system, represented as the set $\mathcal{S} = \{s_1, \ldots, s_N\}$, the system's HLS is defined as the average of the individual scores $s_i$:

$$\text{HLS} = \frac{1}{N} \sum_{i=1}^{N} s_i.$$

We employ HLS to quantify the human-likeness of a TTS system's speech synthesis, which can be assessed both overall and within specific sub-dimensions. The resulting numeric HLS scores can also supervise the training of automated prediction models.

### 3.5 AUTO-ATT

To facilitate rapid evaluation iterations and enhance the usability of the assessment process, we fine-tuned Qwen2.5-Omni-7B (Xu et al., 2025) on a subset of human evaluation data to enable a "model-as-a-judge" approach that allows the model to predict Human-likeness Score (HLS).

**Data.** For training Auto-ATT, we construct a training-testing split from the full ATT corpus at both the corpus and audio levels. At the corpus level, we select three capability subsets—Chinese-English code-switching, character-level pronunciation, and paralinguistics and emotion—as the training corpus, while reserving the remaining two capability subsets for evaluation. On top of this corpus split, we further partition audio by voice: for each of the five model families evaluated in our ATT benchmark (Table 4), we hold out one voice as the test set and use the other three voices for training. To improve the generalization of Auto-ATT, we additionally synthesize speech on the training corpus using internal TTS systems. Specifically, we recruit 437 annotators from crowdsourcing platforms to evaluate all training clips following our protocol, and aggregate labels from three independent annotators per clip into a final label. Details about the corpus and voice splits are provided in Appendix E. During training, each mini-batch is drawn from a single capability subset to maintain subset-level consistency.

**Training.** We utilized TTS-generated speech segments accompanied by instructional prompts designed to guide the model in evaluating speech human-likeness. These inputs were employed to adapt Qwen2.5-Omni-7B for HLS prediction.

Though originally introduced as an auto-regressive audio language model, we adapt Qwen2.5-Omni-7B for HLS score regression by leveraging the logits from its existing `lm_head`. Specifically, we selected three semantically significant tokens: Human, Unclear, and Machine, whose logits represent the model's internal judgments regarding speech quality. A Softmax function was applied to these logits to obtain a normalized probability distribution across the three quality categories. Subsequently, this distribution was converted into a weighted average score by associating each category with a predefined discrete HLS score value: 1 for Human, 0.5 for Unclear, and 0 for Machine. The predicted HLS was calculated as follows:

$$s_i^{\text{pred}} = \sum_{\text{Label}} P(\text{Label}) \cdot [1 \cdot \mathbb{1}(\text{Label} = \text{Human}) + 0.5 \cdot \mathbb{1}(\text{Label} = \text{Unclear})] \tag{1}$$

Logits were specifically extracted from the final token position of the input prompt, denoted by the character "\n". The input prompt comprises both audio content and instructional guidance.

During training, we adopted a loss function consisting of a weighted linear combination of Mean Squared Error (MSE) and Bradley-Terry (BT) (Hunter, 2004) losses:

$$\mathcal{L}_{\text{Total}} = 0.4 \times \mathcal{L}_{\text{BT}} + 0.6 \times \mathcal{L}_{\text{MSE}}, \tag{2}$$

where $\mathcal{L}_{\text{BT}} = -\sum_{(i,j), s.t., s_i^{gt} > s_j^{gt}} \log \sigma \left( s_i^{\text{pred}} - s_j^{\text{pred}} \right)$ and $\mathcal{L}_{\text{MSE}} = \frac{1}{2} \sum_i \left( s_i^{\text{pred}} - s_i^{\text{gt}} \right)^2$.

The model fine-tuning employed Low-Rank Adaptation (LoRA) with hyperparameters configured as follows: rank ($r$) of 32, scaling factor ($\alpha$) of 32, and dropout rate of 0.05. LoRA adapters were applied exclusively to

all linear layers within the LLM component of Qwen2.5-Omni-7B, while other parameters remained fixed throughout the training process.

## 4 EXPERIMENTS

The evaluation involves a total of 20 voice styles across 5 TTS model families including CosyVoice2.0 (Du et al., 2024), MiniMax-Speech (MiniMax, 2025), Seed-TTS (Anastassiou et al., 2024), Step-Audio (Huang et al., 2025) and GPT-4o (Hurst et al., 2024). The voice styles of each model family are detailed in Table 4.

### 4.1 HUMAN EVALUATION

Following the ATT human evaluation protocol outlined in Section 3, we recruited 857 native Chinese speakers through crowdsourcing to evaluate the TTS systems. The participant pool included 202 males, 247 females, and 408 who selected 'Prefer not to say.' As shown in Figure 4, in each evaluation phase, participants will listen to an audio clip and make a single-choice selection afterward, choosing whether the source of audio is [Human] - 1, [Unclear] - 0.5, or [Machine] - 0. Participants were further required to provide written justifications for each of their judgments, which supports a deeper qualitative analysis of the perceptual and decision-making processes underlying their evaluations. Each audio clip took approximately 45 seconds to 1 minute to evaluate and annotate. Compensation was provided at a rate of 0.8 RMB per evaluated clip, equivalent to approximately 48 RMB per hour. To ensure data quality, we applied our predefined validation protocol to screen and verify the collected responses. In addition, we conducted a qualitative coding analysis of the textual justifications, assigning attribution codes to each response. The coding themes and procedural details are described in Appendix B.3. All judgments, justifications, and demographic details were logged anonymously, and the study adhered to the ethical guidelines of the crowdsourcing platform and the researchers' institution.

#### 4.1.1 STATISTICAL SIGNIFICANCE TEST FOR ATT'S HUMAN EVALUATION PROTOCOL DESIGN

To ensure statistical robustness, we conducted a statistical significance test using a Generalized Linear Mixed Model (GLMM) (Bolker et al., 2009). The model showed excellent convergence on the human evaluation data: all parameters had Gelman-Rubin diagnostics ($\hat{R} = 1.00 < 1.01$) and effective sample sizes (ESS $> 400$), indicating precise inference and reliable posterior estimates.

The fixed effects analysis indicates that the mean scores of all evaluated models were statistically significantly higher than the zero baseline (with 95%HDI entirely above zero). Detailed results are provided in Table 2. The findings indicate that the Seed-TTS and Minimax-Speech models significantly outperformed the GPT-4o and CosyVoice models, while the Step-Audio model showed intermediate performance.

Table 2: **Posterior summary statistics from the GLMM**. Including posterior means, standard deviations (SD), 95% highest density intervals (HDI).

| Models | Posterior Mean(SD) | 95%HDI |
|---|---|---|
| Seed-TTS | 0.417 (0.011) | [0.398, 0.438] |
| MiniMax-Speech | 0.387 (0.011) | [0.368, 0.407] |
| Step-Audio | 0.286 (0.011) | [0.266, 0.307] |
| CosyVoice | 0.234 (0.010) | [0.214, 0.254] |
| GPT-4o | 0.138 (0.011) | [0.118, 0.158] |

The random effects analysis reveals significant baseline differences across participants, with the estimated standard deviation of random intercepts being 0.234 (95%HDI = [0.222, 0.246]), suggesting substantial individual variability in overall scoring tendencies. Additionally, there was a moderately positive correlation in repeated evaluations of the same model by individual raters (random slope standard deviation = 0.108, 95%HDI = [0.100, 0.116]), indicating stable preferences or biases in participants' judgments of specific models. We additionally report MOS-based evaluations in Appendix C. The results show strong con-

sistency between HLS and MOS in assessing audio quality. However, the HLS scores exhibit a substantially higher signal-to-noise ratio (Johnson, 2006) (10.53 vs. 5.79 for MOS), indicating greater separability across models and, by implication, a lower annotator burden.

### 4.1.2 BENCHMARKING VIA HUMAN EVALUATION

**Effectiveness of ATT.** As shown in Figure 2, in ATT's benchmark results, Seed-TTS heads the first performance tier with Minimax-Speech. Step-Audio and CosyVoice occupy the second tier with mid-range scores between 0.22 and 0.27, while GPT-4o falls into a distinct third tier at just 0.13, well below the leaders.

The pronounced stepwise gaps show that the ATT evaluation framework can clearly distinguish capability differences among TTS systems. The most notable result is that the highest model's HLS is only 0.4 (Seed-TTS), which remains substantially below the level of true human-likeness. This result markedly deviates from the MOS scores widely reported in prior studies, where TTS systems have often been rated as nearly indistinguishable from human speech. This discrepancy suggests that the HLS metric in the ATT framework is more sensitive and effective in capturing the subtle differences between synthetic and human speech, thereby providing a more realistic assessment of TTS human-likeness.

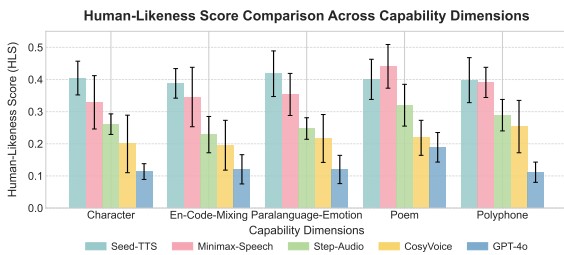

Figure 2: **The Key Benchmark Results of ATT Human Evaluation.**

**Performance of Different Dimensions and Different Voice Styles.** Leveraging ATT's capability for cross-model comparison, we conducted a more fine-grained analysis of the human-likeness exhibited in different voice timbres generated by each TTS system, as well as their overall performance across multiple dimensions. Importantly, as shown in Figure 2 all the models' scores on each sub-dimension mirror their positions in the overall league table, showing no large fluctuations between individual skills and total capability. Notably, substantial variations in voice style are also observed within individual models. For example, Seed-TTS's top-ranked voice, "Skye," scores 0.47, whereas the lowest-ranked voice scores only 0.35. This clear gap shows that ATT can distinguish quality variations between different timbres generated by the same model. The detailed results can be found in Appendix B.4.

**Attribution Analysis.** The qualitative review of the judges' comments reveals common shortcomings across all vendors: (1) prosodic naturalness: intonation patterns often appear abrupt or unnatural, with long sentences delivered in a word-by-word manner and lacking appropriate micro-pauses, making the synthetic origin readily detectable; and (2) expressive richness: emotional expression is either overly flattened or semantically incongruent with the content of the sentence. GPT-4o's Chinese voices are additionally hindered by a noticeable foreign accent, poor rhythm control, and prominent audio artifacts (electronic hiss and noise), which compound its prosodic issues and place it firmly at the bottom.

### 4.2 EFFECTIVENESS OF AUTO-ATT EVALUATION

To validate the effectiveness of Auto-ATT, we design experiments from two aspects: (1) comparing Auto-ATT performance against other MOS-prediction models and (2) measuring Auto-ATT alignment with human judgments.

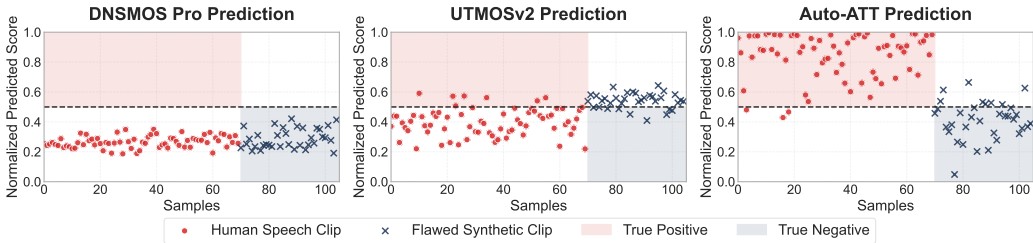

Figure 3: **Trap Items predictions of DNSMOS Pro, UTMOSv2, and Auto-ATT.** For a human speech clip, the ideal outcome is a true positive: the red dot should fall within the red zone; for a flawed synthetic speech clip, the ideal outcome is a true negative: the gray dot should fall within the gray zone.

### 4.2.1 COMPARISON WITH OTHER AUTO EVALUATION IN TRAP ITEM

To evaluate model reliability, we conduct experiments on the trap items included in the ATT-Corpus. We compare the state-of-the-art automatic evaluation methods UTMOSv2 (Baba et al., 2024) and DNSMOS Pro (Cumlin et al., 2024) with our Auto-ATT in predicted HLS on these trap items. Since trap items are readily distinguishable to human listeners in our data validation process, we scored them with each prediction model. These trap items have never been seen by any automatic evaluation methods we evaluated here, so this is a fair comparison. In principle, a reliable model should accurately predict the quality of trap items. For both MOS prediction and HLS scores, human speech should receive significantly higher ratings than defective synthetic speech. As shown in Figure 3, Auto-ATT predicts trap items markedly better than conventional MOS prediction models. Auto-ATT vastly outperformed the baselines, achieving an F1 score of 0.92, while UTMOSv2 reached only 0.14 and DNSMOSPro collapsed to 0.00 at the 0.5 decision threshold. This result indicates that, in comparison to conventional MOS prediction models, Auto-ATT demonstrates superior capability in distinguishing the human-likeness of speech audio, making it particularly well-suited for automated evaluation tasks.

### 4.2.2 CONSISTENCY OF HUMAN EVALUATION

To validate the alignment between Auto-ATT predictions and human assessments, we test Auto-ATT and the base Qwen2.5-Omni-7B on the same audio clips used in our ATT human study, and have both models predict HLS for each capability dimension. This evaluation adopts a strict held-out setting at the voice-style level: for every TTS model family, one voice style is excluded from Auto-ATT's training data and used only for testing. Moreover, the evaluated capability dimensions span both in-distribution subsets seen during training and out-of-distribution subsets held out from training. We aggregate clip-level predicted HLS to obtain voice-level human-likeness scores within each dimension, and measure ranking agreement with human evaluations using PLCC and SRCC. As shown in Table 3, Auto-ATT produces voice rankings that closely track human judgments across all dimensions, achieving near-perfect correlations on in-distribution capabilities and strong alignment on the held-out OOD capabilities, while consistently outperforming Qwen2.5-Omni-7B. To further assess the robustness of Auto-ATT under distributional shift, we additionally evaluate its behavior on entirely unseen TTS system families. Specifically, we apply the ATT courpus to two unseen TTS systems: ElevenLabs Eleven v3 (Staniszewski & Dabkowski, 2025) and Qwen3-TTS-Flash (Qwen Team, 2025), and compare Auto-ATT's voice-level rankings with human judgments

Table 3: SRCC and PLCC of Auto-ATT and Qwen2.5-Omni-7B across different capability dimensions.

| Capability Dimension | Auto-ATT | Qwen2.5 Omni |
|---|---|---|
| Metrics | SRCC / PLCC | SRCC / PLCC |
| *In-Distribution Dimensions* | | |
| Special Characters and Numerals | **1.00 / 0.949** | 0.899 / 0.708 |
| Chinese-English Code-switching | **1.00 / 0.945** | 0.899 / 0.811 |
| Paralinguistic Features and Emotions | **0.899 / 0.933** | 0.700 / 0.677 |
| *Out-of-Distribution Dimensions* | | |
| Classical Chinese Poetry/Prose | **0.899 / 0.916** | 0.600 / 0.571 |
| Polyphonic Characters | **0.899 / 0.889** | 0.499 / 0.725 |

on their synthesized audio. The experimental details can be found in Appendix E.2. Despite substantial differences from the families used in Auto-ATT's training, the model continues to exhibit strong agreement with human assessments on the held-out OOD capability dimensions. Auto-ATT attains SRCC / PLCC scores of 0.714 / 0.886 on *Classical Chinese Poetry/Prose* and 0.771 / 0.790 on *Polyphonic Characters*. These results indicate that Auto-ATT serves as a reliable proxy for human-likeness assessment, with robust generalization to different voice styles, unseen TTS systems and even unseen capability criteria.

## 5 CONCLUSION & LIMITATIONS

In this paper, we propose the Audio Turing Test (ATT), an innovative evaluation framework specifically designed to address critical challenges in evaluating the human-likeness of LLM-based TTS systems in Chinese. ATT uniquely integrates a comprehensive, multi-dimensional evaluation corpus ATT-Corpus with a robust Turing-Test-inspired evaluation protocol, thereby providing both qualitative and quantitative insights. Our rigorous validation demonstrates that ATT reliably differentiates among state-of-the-art LLM-based TTS models, pinpointing specific strengths and weaknesses across diverse linguistic dimensions such as code-switching, emotional expression, polyphony, and classical texts. Additionally, by finetuning Qwen2.5-Omni-7B on human annotations, we develop Auto-ATT for accelerating the iteration cycles of TTS systems through rapid and accurate assessments. Results confirm Auto-ATT's superior alignment with human evaluators compared to traditional automatic evaluation methodologies. A current limitation of ATT is its language-specific nature, as both the protocol and corpus are primarily designed for Chinese speech synthesis. To address this, we aim to extend the ATT framework to support multiple languages and a broader range of speech synthesis scenarios, thereby validating its generalizability and cross-linguistic effectiveness. Overall, ATT represents a significant advancement in the evaluation of LLM-based speech synthesis systems and paves the way for more natural and human-like TTS technologies.

**The Use of Large Language Models.** We used a large language model as a general-purpose assistant solely for text editing, including grammar correction, wording and tone adjustments, punctuation, and stylistic consistency. The model did not contribute to research ideation, methodology, experimental design, data analysis, interpretation of results, or the generation of substantive academic content or references. All suggestions were reviewed and approved by the authors, who take full responsibility for the final text. Our use of LLMs for data synthesis/augmentation is described in the main manuscript; this statement pertains only to editorial assistance.

**Ethics Statement.** Our method and algorithm do not involve any adversarial attack, and will not endanger human security. All our experiments does not involve ethical and fair issues.

**Reproducibility Statement.** The ATT-Corpus is available at supplementary materials, and we will release our Auto-ATT model and code in huggingface once the paper being accepted. We specify all the implementation details of our methods in Appendix B. The experiment additional results are in the Appendix B.4 and Appendix D.

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

# A ATT-CORPUS DETAILS

## A.1 DATA GENERATION

We found that we could not directly synthesize colloquial texts that met our requirements, so we designed a three-step corpus-creation workflow: 1) use GPT-4o (Hurst et al., 2024) to batch-generate Chinese sentences that mix in English, 2) pass these sentences through DeepSeek-R1 (Guo et al., 2025) for a colloquial adapt, 3) have linguistics experts further enrich and diversify the text through rewriting and perform final verification.

**Batch Generate.** We first employed GPT-4o (Hurst et al., 2024) to generate texts tailored to each predefined capability dimension. For example, for the Chinese-English code-switching dimension, we began by using the following prompt to produce Chinese sentences that incorporate English words.

---

给我一些日常沟通的的中文长文本，每一句话中需要有非常自然的中英文掺杂的现象，一句话只出现1-2个单词，且主要为专有词汇，或英文的filler words。

示例一：今天在朋友圈看到朋友发的自拍，她在用一个叫FaceTune的app修图，效果真的是很棒，很自然，你要不要也试试？

示例二：昨晚在Hulu上看了一部新的浪漫喜剧，叫《To All the Boys I've Loved Before》，剧情特别甜，看完之后觉得心情特别好。

示例三：今天在星巴克点了一杯新的Cold Brew Coffee，味道特别醇厚，喝完感觉一整天都特别清醒，推荐你也试试，很提神哦！

示例四：最近我一直在用 Estée Lauder 的粉底液，它的妆效很 natural，能够很好地贴合肌肤，遮盖瑕疵的同时又不会显得很厚重，让我的肌肤看起来自然无瑕，仿佛天生丽质一般。

示例五：今天我在网易云音乐上闲逛的时候，发现了一首超好听的新歌，叫《Shape of You》。那旋律可动感了，我听完之后，心情瞬间变得超好，感觉整个人都跟着节奏摇摆起来了，你听过这首歌吗？

执行后将每句话的长度拓展到100字左右。执行后将部分句子的句末加一些语气助词，丰富句子的口语化程度，但不要夸张。需要熟悉中国人的口语习惯，然后生成以上要求内容。请给我40句

---

**Colloquial Rewrite.** To make the text still more conversational, we ran it through DeepSeek-R1 (Guo et al., 2025) for an additional colloquial rewrite, using the prompt shown below:

---

将给出的文本改写为更加口语化，有沟通感的文本，并添加一定的背景及前后连贯信息，你可以从以下的6个示例中获得灵感，但不允许照搬照抄，或者仿照句式，不允许用同样重复的开头

示例1：原始：开始用Notion这个app之后，发现它真的太强大了，不仅可以用来记笔记，还能用来管理项目和计划，非常实用，简直是提高效率的利器呢。更改为：我跟你讲，我最近在用Notion这个app，我的天我真的发现它真的很强，不仅可以用来记笔记，还能用来管理项目和计划，而且还很美观，真的挺实用的，是个提高效率的好东西，你们要不要也用一下看看？

示例2：原始：朋友推荐了一个新的K-pop组合，叫BTS，听了他们的几首歌后真的觉得很好听，特别是那首《Dynamite》，旋律超级洗脑，推荐你也去听听看。更改为：昨天跟家里那帮朋友出去吃饭，他们给我推荐了一个新出来的的K-pop组合，叫BTS，还挺不错，听了他们的几首歌都还

---

可以，特别《Dynamite》这首，旋律超级洗脑，我从吃饭一直哼到回家洗澡，睡觉的时候脑子里都还在放这首歌，没救了。

示例3：原始：下载了Pocket这个app，用来保存平时看到的好文章，觉得特别方便，这样有时间的时候就可以慢慢看，不会错过任何好内容，真的是读书神器。更改为：天，哥们儿我跟你说，昨天我刚下载了Pocket这个app，发现它可以把平时看到的好文章都保存下来，也太方便了吧！你要不也用用看？这样有时间的时候就可以慢慢看，就不用担心错过很多不错的内容啦，真是读书神器绝绝子，安利你！

示例4：原始：最近迷上了刷TikTok，真的有好多搞笑的短视频，看得我笑到不行，特别是那些创意短视频，简直让人一刷就停不下来，你也常常刷TikTok吗？更改为：哇塞真的，TikTok一刷就停不下来，真的好多视频贼搞笑，短小精悍，看得我笑到不行！发明这些创意视频的博主也太有才了吧，好多时间一看就一两个小时过去了，你也刷TikTok吗？咱加个好友不。

示例5：原始：昨晚在Hulu上看了一部新电影，叫《寄生虫》，剧情超精彩，每个情节都有出人意料的反转，看得我完全停不下来，一口气看完了整部电影，特别推荐。更改为：你有看过最近大火的新电影《寄生虫》吗？我昨天在Hulu上看的，剧情好精彩啊，每个情节的反转都特别出人意料，根本想不到接下来会发生什么。其实我随手点开的，没想到会越看越上瘾，完全停不下来，最后一口气看完了，我跟你说你一定要去看，看完了记得和我分享。

示例6：原始：开始用Headspace做冥想，每天花十分钟，整体状态变好了很多，特别是它的音指导很温柔，特别容易进入冥想状态，感觉整个人都特别放松。更改为：最近不知道怎么了精神状态很差，所以我跟着一个叫Headspace的节目做冥想，每天花十分钟放空自己，练了快一个月，感觉自己压力没那么大了，睡眠质量也更好了，说起来我觉得这个channel最棒的是声音指导很温柔，你听了那个声音就很容易进入冥想状态，就觉得整个人好像在泡澡一样，特别安稳。

---

## A.2 AUDIO DATA VALIDATION CRITERIA

**Synthesis Success.** Synthesis success refers to the correctness of the output audio in terms of overall audio quality, transcription accuracy, and language appropriateness. Specifically, we check for issues such as: significant audio quality defects (e.g., excessive robotic noise, jittering), extremely short or incomplete audio that cannot be properly transcribed (e.g., only a single "ah" sound or complete silence), language mismatch (e.g., input in Chinese but output in Japanese), inconsistencies in voice timbre within a single clip (e.g., mixing multiple voice styles), and other cases where the output is unintelligible in the target language.

**Synthesis Consistency.** Synthesis consistency refers to the consistency of output when the same text is synthesized multiple times using the same voice and technology. This assessment focuses on whether the resulting audio clips are consistent in overall characteristics such as voice timbre (e.g., gender, age), language (e.g., remaining within the same language such as Chinese or English), and prosody (e.g., intonation, stress, and tone of voice). The goal is to determine whether the outputs can reliably be attributed to the same voice.

## A.3 BLACK-BOX AND WHITE-BOX.

To ensure a fair and reliable evaluation, we divide the generated data into white-box and black-box subsets. The white-box subset is made publicly available, while the black-box subset is hosted on an evaluation platform for open and blind testing. Our experiments validate the consistency between white-box and black-box evaluation results.

Table 4: **The model families and their voice styles we evaluated.**

| Model Families | Voice Styles |
|---|---|
| CosyVoice2.0 (Du et al., 2024) | longshuo, longxiaocheng, longxiaochun, longxiaoxia |
| MiniMax-Speech (MiniMax, 2025) | xinyue, yaoyao, siyuan, zixuan |
| Seed-TTS (Anastassiou et al., 2024) | Skye (zh_female_shuangkuaisisi_moon_bigtt), Alvin (zh_male_wennuanahu_moon_bigtts), Brayan (zh_male_shaonianzixin_moon_bigtts), Moon (zh_female_linjianvhai_moon_bigtts) |
| Step-Audio (Huang et al., 2025) | qingniandaxuesheng, shenchennanyin, linjiajiejie, wenjingxuejie |
| GPT-4o (Hurst et al., 2024) | Alloy, Shimmer, Echo, Onyx |

# B  ATT BENCHMARK DETAILS

## B.1  EVALUATED TTS SYSTEMS

Seed-TTS (Anastassiou et al., 2024) is ByteDance's large-scale foundation family for speech generation-its flagship autoregressive language-model variant scales into the multi-billion-parameter range and is trained with data and model sizes "orders of magnitude larger" than previous TTS systems, plus an optional diffusion decoder Seed-TTS-DiT. Seed-TTS offers zero-shot speaker cloning, fine-grained emotion control and in-context speech editing while matching human naturalness scores in CMOS.

MiniMax-Speech-01 (MiniMax, 2025) is an autoregressive Transformer TTS with an integrated learnable speaker encoder that enables true zero-shot voice cloning across 32 languages. Although its exact size is undisclosed, the model is built on the same infrastructure as MiniMax-Text-01 (456B total/45.9B active parameters), so it inherits Mixture-of-Experts efficiency and ultra-long-context techniques from that 456B-parameter backbone.

CosyVoice2.0 (Du et al., 2024) delivers sub-150 ms first-packet latency in both streaming and offline modes, with multilingual zero-shot voice cloning across Chinese, English, Japanese, Korean and many dialects. Public checkpoints of CosyVoice2.0 range from 300 M to 0.5 B parameters.

Step-Audio (Huang et al., 2025) pairs a 130 B-parameter multimodal generative engine that synthetically bootstraps training data with a resource-efficient 3 B-parameter Step-Audio-TTS synthesiser. This combination supports controllable speech with emotions, dialects and styles, and meets real-time requirements through speculative decoding and a dual-codebook tokenizer architecture.

OpenAI's GPT-4o (Hurst et al., 2024) is an end-to-end multimodal model (parameter count not publicly disclosed) that handles text, vision and audio in one network and speaks with human-like latency-232 ms best-case, 320 ms on average. It matches GPT-4-Turbo on text but adds expressive speech synthesis, real-time translation and paralinguistic cues without the separate ASR and TTS stages used in previous Voice Mode pipelines.

## B.2  INSTRUCTIONS AND USER INTERFACE

We provide instructions for each participant for the evaluation task and design the reward system to encourage the high-quality evaluation.

Since our benchmark are in Chinese, our instructions are also in Chinese for native speaker participants. Here we provide a translated English version for review:

```
Task description

In this task, you must decide whether each audio clip you hear is spoken by a real person or generated
 by a machine, and you must state why you reached that conclusion.
```

```
Your written reason is the main evidence used in manual review, so base it on concrete observations of
 the recording.

For every 10 clips there are several hidden "test items."
These have an unmistakably correct answer; selecting the wrong answer on a test item will cause your
entire submission to fail review. Do not rely on AI to draft your responses-judgements that fail
review will be discarded and not counted as valid data.

How to write your reason

Examples of poor reasons

(Not convincing; give no specific evidence from the audio)
     1.      "Pure machine voice."
     2.      "The imitation of human speech is too forced."
     3.      "Obviously a machine tone-doesn't sound like a real person."
     4.      "Sounds like a late-night radio host."

Examples of good reasons

(Accurate analysis that cites concrete details in the clip)
     1.      The phrase "Many thins" should end with a falling intonation, but here it rises-it
     sounds unnatural.
     2.      The clip is machine-generated: each word pops out individually with poor flow.
     3.      The phrase "go away" lacks the angry/impatient tone that should be present.
     4.      After the word "angry," the breath has a noticeable electronic/robotic artifact.
```

And the user interface for the task are shown in Figure 4 with explanation in English.

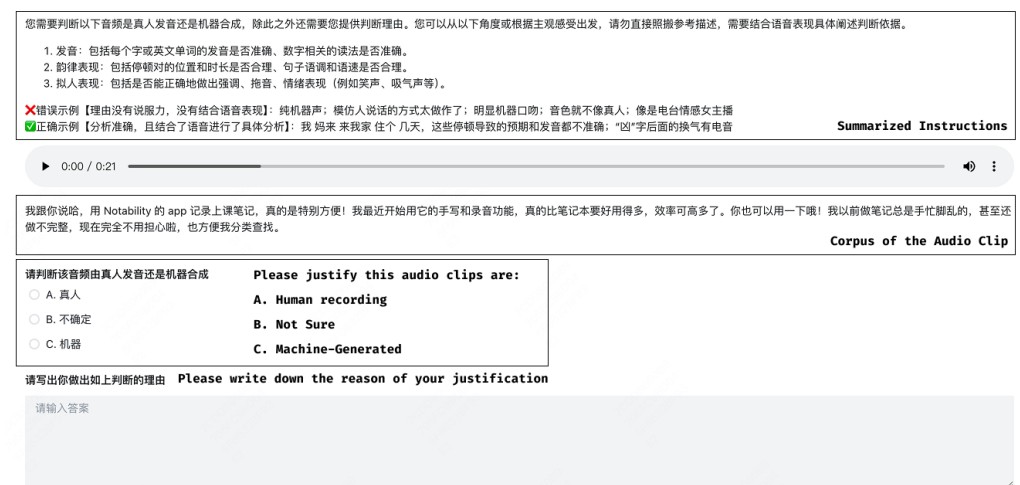

Figure 4: The Screen of One Audio Clip in ATT Evaluation.

### B.3 QUALITATIVE ANALYSIS

The coding criteria for qualitative analysis are based on Table 5, which consists of four dimensions: first, pronunciation accuracy, focusing on the correctness of each Chinese character's pronunciation (especially polyphonic characters within words), accuracy of tones, correct pronunciation of embedded English words, and accurate pronunciation of numerical information such as dates, monetary amounts, and phone numbers;

second, prosodic appropriateness, examining whether pauses occur at reasonable positions with appropriate duration, whether the sentence intonation matches semantic intentions (e.g., questions or exclamations), and whether speech speed is appropriate without being excessively fast or slow; third, audio clarity, assessing overall audio quality, including the presence of noticeable background noise, jitter, or electronic distortion in pronunciations; and fourth, naturalness and human-like expressiveness, evaluating whether the overall speech performance appears human-like and natural by considering factors such as semantic emphasis and prolongation of words, emotional expressions consistent with sentence meaning, and effective paralinguistic features including breaths, laughter, crying, coughing, or breathy voice.

Table 5: Criteria for Qualitative Analysis

| Dimension | Detailed Explanation |
|---|---|
| Pronunciation Accuracy | - Whether each Chinese character is pronounced correctly, especially polyphonic characters within words. - Whether the tones of characters/words are accurate. - Whether embedded English words are pronounced correctly. - Whether numerical information such as dates, monetary amounts, and phone numbers is read accurately. |
| Prosodic Appropriateness | - Whether the position and duration of pauses are reasonable. - Whether the intonation matches the sentence meaning, such as questions or exclamations. - Whether speech speed is appropriate, avoiding overly fast or slow pacing. |
| Audio Clarity | - Whether the overall audio quality is clear, and if noticeable background noise is present. - Whether pronunciations have jitter, electronic distortion, or other clarity issues. |
| Naturalness and Human-like Expressiveness | - Whether the overall speech appears natural and comparable to human speech, considering:
• Appropriate semantic emphasis on words.
• Appropriate prolongation of words matching semantic context.
• Emotional expressions matching the sentence context.
• Effective use of paralinguistic features such as breathing sounds, laughter, crying, coughing, or breathy voice. |

## B.4 DETAIL RESULTS

**Soundness of the black-/white-box split.** Crucially, the overall performance hierarchy remains consistent when comparing white-box and black-box evaluation settings: each model retains the same relative ranking across both conditions (as shown in Figure 2). The small and uniform performance gap between the two settings indicates that they are of comparable difficulty, confirming that the black-box/white-box split is well-balanced and does not introduce systematic bias into the evaluation.

**Dimensional Performance.** Across ATT's five evaluation dimensions, Seed-TTS consistently ranks first, demonstrating the strongest overall performance and particularly excelling at Chinese-English Code-switching and Special Characters and Numerals; its only relative weakness is in Classical Chinese Poetry/Prose, where it is narrowly outperformed by Minimax-Speech. Step-Audio, CosyVoice, and GPT-4o follow in that order.

**Different Voice Styles Performance.** We list the performance of each voice style in Table 6.

### B.5 HUMAN LABEL STATISTICS

To examine whether our evaluation could be biased by participants overusing the [Unclear] option, we analyze the annotator-level unclear rate, i.e., the fraction of instances an annotator marked as [Unclear] among all instances they labeled.

Overall, the use of [Unclear] is low and highly concentrated among a small subset of annotators. Among the 857 annotators in our evaluation set, 565 annotators (65.93%) never selected [Unclear] at all. 655 annotators (76.43%) have an unclear rate no more than 5%, and 728 annotators (84.95%) have an unclear rate no more than 10%. Only 93 annotators (10.85%) fall into the 10%–30% range, and merely 36 annotators (4.20%) exceed 30%. Consistently, the median unclear rate across annotators is 0.00%, with a mean of 4.62% and a standard deviation of 9.56%, indicating a right-skewed but overall low usage pattern.

These statistics show that [Unclear] was not a dominant choice during labeling; most annotators provided decisive labels for nearly all evaluation instances. Therefore, our reported evaluation results are not driven by widespread avoidance via [Unclear], but rather reflect performance on clearly judged samples.

## C COMPARISON WITH MEAN OPINION SCORE RESULTS

Table 7 reports the posterior analysis of the MOS benchmark. In the human study, participants rated audio quality on a 5-point scale. For ease of comparison, the scores in Table 7 are linearly normalized to $[0, 1]$.

## D AUTO-ATT EXPERIMENTS ADDITIONAL RESULTS

We used 4 NVIDIA A100 GPUs to train Auto-ATT, which takes about 1 hour. The server's CPU was an Intel Xeon Platinum 8358P (2.60 GHz, 128 cores). Table 8 and Table 9 present detailed Auto-ATT evaluation results for both white-box and black-box scenarios.

## E ADDITIONAL AUTO-ATT DETAILS

### E.1 DATA SPLIT DETAILS

We detail the voice-level training–testing partition in this appendix. For each of the five model families in Table 4, we hold out exactly one voice style to form the Auto-ATT test set, and use the remaining three voices from the same family for training. Concretely, the held-out test voices are: longxiaochun (CosyVoice2.0), Moon (Seed-TTS), siyuan (MiniMax-Speech), Echo (GPT-4o), and shenchennanyin (Step-Audio). This split ensures that Auto-ATT is evaluated on unseen voices within each family.

### E.2 UNSEEN TTS SYSTEMS RESULTS

To test whether our ATT corpus and Auto-ATT pipeline can be directly applied to newly released TTS systems, we run an additional evaluation on two unseen model families that were not part of the original benchmark or

Table 6: HLS of Different Voice Styles with 95% Confidence Interval

| Model | Voice Style | Special Characters and Numerals | Chinese-English Code-switching | Paralinguistic Features and Emotions | Classical Chinese Poetry/Prose | Polyphonic Characters |
|---|---|---|---|---|---|---|
| CosyVoice | longshuo | 0.108 [0.035, 0.190] | 0.118 [0.066, 0.194] | 0.135 [0.058, 0.221] | 0.180 [0.052, 0.324] | 0.170 [0.100, 0.249] |
| | longxiaocheng | 0.211 [0.106, 0.325] | 0.128 [0.028, 0.247] | 0.140 [0.066, 0.220] | 0.188 [0.110, 0.260] | 0.198 [0.087, 0.313] |
| | longxiaochun | 0.125 [0.054, 0.199] | 0.233 [0.126, 0.345] | 0.262 [0.170, 0.356] | 0.222 [0.093, 0.359] | 0.263 [0.123, 0.411] |
| | longxiaoxia | 0.355 [0.248, 0.464] | 0.305 [0.165, 0.453] | 0.330 [0.237, 0.423] | 0.285 [0.185, 0.394] | 0.385 [0.274, 0.502] |
| MiniMax-Speech | siyuan | 0.278 [0.203, 0.351] | 0.313 [0.156, 0.479] | 0.308 [0.160, 0.462] | 0.365 [0.179, 0.560] | 0.329 [0.224, 0.435] |
| | xinyue | 0.458 [0.348, 0.568] | 0.417 [0.264, 0.573] | 0.450 [0.328, 0.569] | 0.515 [0.394, 0.636] | 0.433 [0.318, 0.547] |
| | yaoyao | 0.363 [0.299, 0.491] | 0.428 [0.265, 0.592] | 0.350 [0.234, 0.468] | 0.455 [0.331, 0.583] | 0.428 [0.295, 0.563] |
| | zixuan | 0.218 [0.119, 0.323] | 0.225 [0.101, 0.360] | 0.308 [0.170, 0.453] | 0.430 [0.324, 0.516] | 0.375 [0.238, 0.422] |
| Seed-TTS | Alvin | 0.400 [0.286, 0.516] | 0.360 [0.210, 0.514] | 0.395 [0.237, 0.555] | 0.400 [0.256, 0.546] | 0.363 [0.242, 0.485] |
| | Brayan | 0.413 [0.351, 0.476] | 0.393 [0.262, 0.526] | 0.360 [0.224, 0.500] | 0.405 [0.265, 0.545] | 0.430 [0.329, 0.532] |
| | moon | 0.365 [0.238, 0.497] | 0.360 [0.257, 0.463] | 0.400 [0.244, 0.561] | 0.323 [0.207, 0.383] | 0.300 [0.179, 0.426] |
| | skye | 0.440 [0.256, 0.629] | 0.440 [0.309, 0.572] | 0.518 [0.392, 0.643] | 0.475 [0.328, 0.622] | 0.500 [0.386, 0.612] |
| GPT-4o | alloy | 0.153 [0.074, 0.237] | 0.095 [0.021, 0.183] | 0.155 [0.083, 0.232] | 0.171 [0.097, 0.250] | 0.101 [0.019, 0.204] |
| | echo | 0.085 [0.028, 0.143] | 0.098 [0.023, 0.182] | 0.056 [0.004, 0.120] | 0.143 [0.023, 0.289] | 0.075 [0.009, 0.158] |
| | onyx | 0.100 [0.044, 0.158] | 0.135 [0.037, 0.246] | 0.095 [0.025, 0.175] | 0.196 [0.101, 0.293] | 0.120 [0.052, 0.178] |
| | shimmer | 0.118 [0.069, 0.201] | 0.155 [0.032, 0.306] | 0.175 [0.096, 0.255] | 0.246 [0.163, 0.332] | 0.150 [0.082, 0.218] |
| Step-Audio | wenjingxuejie | 0.233 [0.135, 0.341] | 0.252 [0.106, 0.422] | 0.238 [0.116, 0.372] | 0.363 [0.251, 0.429] | 0.329 [0.230, 0.425] |
| | shenchenmanyin | 0.243 [0.170, 0.386] | 0.214 [0.101, 0.345] | 0.258 [0.166, 0.352] | 0.243 [0.151, 0.309] | 0.283 [0.184, 0.386] |
| | linjiajiejie | 0.266 [0.181, 0.406] | 0.181 [0.103, 0.262] | 0.210 [0.117, 0.305] | 0.270 [0.139, 0.410] | 0.213 [0.106, 0.329] |
| | qingniandaxuesheng | 0.304 [0.195, 0.424] | 0.268 [0.124, 0.434] | 0.285 [0.190, 0.360] | 0.405 [0.331, 0.519] | 0.332 [0.202, 0.379] |

Table 7: **Posterior summary statistics of Mean Opinion Score from the GLMM**. Including posterior means, standard deviations (SD), 95% highest density intervals (HDI).

| Models | Posterior Mean(SD) | 95%HDI |
|---|---|---|
| Seed-TTS | 0.680 (0.020) | [0.650, 0.710] |
| MiniMax-Speech | 0.620 (0.020) | [0.590, 0.650] |
| Step-Audio | 0.560 (0.020) | [0.530, 0.590] |
| CosyVoice | 0.470 (0.020) | [0.440, 0.490] |
| GPT-4o | 0.390 (0.020) | [0.360, 0.420] |

Auto-ATT training: ElevenLabs Eleven v3 (Staniszewski & Dabkowski, 2025) and Qwen3-TTS-Flash (Qwen Team, 2025). We follow the same procedure as in the main study. Specifically, for each unseen family we collect audio outputs for the full ATT prompt set using their official voice styles (ElevenLabs: Chris, Matilda, Sarah, Will; Qwen3-TTS-Flash: Cherry, Ethan). We then conduct human evaluation on these clips using the same annotation protocol, and aggregate clip-level scores to voice-level HLS. Table 10 reports the posterior mean HLS with 95% confidence intervals for each capability dimension.

These human results provide a realistic snapshot of unseen-family performance under the ATT benchmark and serve as the basis for assessing Auto-ATT's applicability to newly emerged TTS systems.

Table 8: HLS of Different Voice Styles with 95% Confidence Interval in White-box Corpus

| Model | Voice Style | Special Characters and Numerals | Chinese-English Code-switching | Paralinguistic Features and Emotions | Classical Chinese Poetry/Prose | Polyphonic Characters |
|---|---|---|---|---|---|---|
| CosyVoice | longshuo | 0.041 [0.029, 0.052] | 0.036 [0.027, 0.046] | 0.026 [0.017, 0.034] | 0.009 [0.005, 0.012] | 0.013 [0.009, 0.017] |
| | longxiaocheng | 0.014 [0.009, 0.018] | 0.022 [0.015, 0.029] | 0.016 [0.008, 0.024] | 0.015 [0.010, 0.021] | 0.015 [0.008, 0.022] |
| | longxiaochun | 0.126 [0.102, 0.150] | 0.093 [0.075, 0.112] | 0.100 [0.079, 0.120] | 0.008 [0.006, 0.010] | 0.032 [0.022, 0.041] |
| | longxiaoxia | 0.311 [0.279, 0.342] | 0.252 [0.225, 0.280] | 0.293 [0.261, 0.326] | 0.030 [0.018, 0.041] | 0.108 [0.085, 0.130] |
| Seed-TTS | Alvin | 0.334 [0.309, 0.359] | 0.336 [0.313, 0.359] | 0.313 [0.287, 0.338] | 0.118 [0.098, 0.139] | 0.201 [0.174, 0.227] |
| | Brayan | 0.457 [0.440, 0.475] | 0.460 [0.447, 0.474] | 0.388 [0.368, 0.408] | 0.174 [0.149, 0.199] | 0.348 [0.325, 0.371] |
| | moon | 0.408 [0.391, 0.425] | 0.393 [0.376, 0.409] | 0.386 [0.367, 0.405] | 0.109 [0.090, 0.129] | 0.215 [0.190, 0.240] |
| | skye | 0.518 [0.504, 0.531] | 0.497 [0.483, 0.511] | 0.516 [0.496, 0.535] | 0.315 [0.286, 0.343] | 0.423 [0.402, 0.445] |
| GPT-4o | alloy | 0.297 [0.271, 0.324] | 0.354 [0.334, 0.375] | 0.237 [0.211, 0.262] | 0.048 [0.037, 0.059] | 0.096 [0.078, 0.114] |
| | echo | 0.206 [0.179, 0.233] | 0.314 [0.288, 0.340] | 0.145 [0.123, 0.167] | 0.024 [0.019, 0.030] | 0.054 [0.042, 0.066] |
| | onyx | 0.264 [0.239, 0.290] | 0.324 [0.301, 0.347] | 0.222 [0.196, 0.247] | 0.053 [0.037, 0.069] | 0.082 [0.064, 0.100] |
| | shimmer | 0.256 [0.231, 0.280] | 0.332 [0.307, 0.357] | 0.181 [0.155, 0.207] | 0.043 [0.031, 0.055] | 0.086 [0.068, 0.104] |
| Minimax-Speech | siyuan | 0.064 [0.048, 0.079] | 0.090 [0.073, 0.108] | 0.067 [0.051, 0.082] | 0.015 [0.010, 0.019] | 0.029 [0.020, 0.037] |
| | xinyue | 0.303 [0.282, 0.325] | 0.309 [0.287, 0.331] | 0.266 [0.243, 0.290] | 0.054 [0.042, 0.067] | 0.132 [0.113, 0.151] |
| | yaoyao | 0.300 [0.280, 0.321] | 0.308 [0.289, 0.328] | 0.261 [0.239, 0.283] | 0.031 [0.022, 0.040] | 0.070 [0.055, 0.085] |
| | zixuan | 0.037 [0.026, 0.049] | 0.077 [0.059, 0.095] | 0.023 [0.015, 0.031] | 0.011 [0.008, 0.014] | 0.016 [0.011, 0.020] |
| Step-Audio | wenjingxuejie | 0.256 [0.217, 0.297] | 0.269 [0.242, 0.298] | 0.193 [0.164, 0.220] | 0.031 [0.021, 0.041] | 0.098 [0.076, 0.120] |
| | shenchenmanyin | 0.040 [0.024, 0.055] | 0.037 [0.025, 0.049] | 0.019 [0.014, 0.024] | 0.011 [0.008, 0.014] | 0.018 [0.013, 0.023] |
| | linjiajiejie | 0.194 [0.159, 0.229] | 0.201 [0.173, 0.228] | 0.107 [0.086, 0.128] | 0.011 [0.008, 0.015] | 0.057 [0.041, 0.072] |
| | qingniandaxuesheng | 0.244 [0.204, 0.283] | 0.231 [0.200, 0.262] | 0.164 [0.137, 0.192] | 0.025 [0.018, 0.033] | 0.069 [0.050, 0.088] |

Table 9: HLS of Different Voice Styles with 95% Confidence Interval in Black-box Corpus

| Model | Voice Style | Special Characters and Numerals | Chinese-English Code-switching | Paralinguistic Features and Emotions | Classical Chinese Poetry/Prose | Polyphonic Characters |
|---|---|---|---|---|---|---|
| CosyVoice | longshuo | 0.087 [0.067, 0.106] | 0.069 [0.053, 0.083] | 0.116 [0.088, 0.144] | 0.007 [0.005, 0.009] | 0.012 [0.008, 0.015] |
| | longxiaocheng | 0.040 [0.027, 0.053] | 0.036 [0.024, 0.049] | 0.047 [0.031, 0.063] | 0.010 [0.007, 0.012] | 0.016 [0.009, 0.023] |
| | longxiaochun | 0.168 [0.142, 0.195] | 0.183 [0.159, 0.208] | 0.196 [0.166, 0.226] | 0.008 [0.006, 0.010] | 0.029 [0.020, 0.039] |
| | longxiaoxia | 0.333 [0.302, 0.362] | 0.348 [0.321, 0.375] | 0.353 [0.316, 0.389] | 0.019 [0.008, 0.029] | 0.108 [0.083, 0.133] |
| Seed-TTS | Alvin | 0.328 [0.302, 0.352] | 0.386 [0.367, 0.405] | 0.290 [0.261, 0.320] | 0.105 [0.085, 0.124] | 0.213 [0.186, 0.239] |
| | Brayan | 0.424 [0.407, 0.441] | 0.473 [0.459, 0.486] | 0.359 [0.332, 0.386] | 0.149 [0.126, 0.171] | 0.325 [0.297, 0.353] |
| | moon | 0.418 [0.401, 0.434] | 0.460 [0.446, 0.475] | 0.388 [0.366, 0.411] | 0.083 [0.065, 0.101] | 0.205 [0.179, 0.231] |
| | skye | 0.506 [0.491, 0.521] | 0.532 [0.521, 0.544] | 0.544 [0.520, 0.567] | 0.244 [0.218, 0.270] | 0.397 [0.376, 0.419] |
| GPT-4o | alloy | 0.297 [0.273, 0.322] | 0.384 [0.363, 0.405] | 0.245 [0.214, 0.276] | 0.036 [0.026, 0.046] | 0.100 [0.080, 0.119] |
| | echo | 0.235 [0.212, 0.258] | 0.334 [0.309, 0.359] | 0.192 [0.163, 0.221] | 0.019 [0.013, 0.024] | 0.060 [0.046, 0.076] |
| | onyx | 0.282 [0.257, 0.307] | 0.367 [0.344, 0.389] | 0.229 [0.197, 0.260] | 0.033 [0.024, 0.043] | 0.089 [0.073, 0.106] |
| | shimmer | 0.271 [0.248, 0.296] | 0.354 [0.330, 0.378] | 0.225 [0.195, 0.255] | 0.035 [0.023, 0.047] | 0.086 [0.068, 0.105] |
| Minimax-Speech | siyuan | 0.171 [0.147, 0.196] | 0.173 [0.149, 0.198] | 0.173 [0.143, 0.203] | 0.017 [0.011, 0.023] | 0.034 [0.025, 0.043] |
| | xinyue | 0.300 [0.278, 0.323] | 0.361 [0.343, 0.378] | 0.298 [0.270, 0.326] | 0.044 [0.033, 0.055] | 0.147 [0.125, 0.169] |
| | yaoyao | 0.334 [0.311, 0.356] | 0.388 [0.372, 0.405] | 0.326 [0.297, 0.355] | 0.025 [0.017, 0.033] | 0.086 [0.071, 0.102] |
| | zixuan | 0.072 [0.051, 0.091] | 0.091 [0.073, 0.109] | 0.064 [0.044, 0.083] | 0.010 [0.007, 0.013] | 0.018 [0.012, 0.023] |
| Step-Audio | wenjingxuejie | 0.272 [0.244, 0.301] | 0.339 [0.312, 0.365] | 0.259 [0.223, 0.295] | 0.027 [0.018, 0.036] | 0.101 [0.080, 0.122] |
| | shenchenmanyin | 0.037 [0.025, 0.048] | 0.066 [0.048, 0.084] | 0.046 [0.031, 0.061] | 0.013 [0.009, 0.017] | 0.017 [0.012, 0.021] |
| | linjiajiejie | 0.195 [0.168, 0.221] | 0.272 [0.245, 0.299] | 0.168 [0.137, 0.200] | 0.010 [0.005, 0.014] | 0.053 [0.039, 0.067] |
| | qingniandaxuesheng | 0.234 [0.205, 0.262] | 0.320 [0.293, 0.347] | 0.178 [0.144, 0.213] | 0.023 [0.015, 0.032] | 0.059 [0.044, 0.075] |

Table 10: HLS of Unseen TTS Systems with 95% Confidence Interval

| Model | Voice Style | Special Characters and Numerals | Chinese-English Code-switching | Paralinguistic Features and Emotions | Classical Chinese Poetry/Prose | Polyphonic Characters |
|---|---|---|---|---|---|---|
| Eleven v3 | Chris | 0.299 [0.238, 0.361] | 0.302 [0.242, 0.361] | 0.318 [0.256, 0.380] | 0.528 [0.460, 0.597] | 0.457 [0.389, 0.524] |
| | Matilda | 0.169 [0.119, 0.219] | 0.126 [0.082, 0.170] | 0.130 [0.085, 0.175] | 0.290 [0.229, 0.351] | 0.179 [0.128, 0.229] |
| | Sarah | 0.191 [0.138, 0.244] | 0.280 [0.221, 0.340] | 0.212 [0.158, 0.266] | 0.472 [0.404, 0.541] | 0.276 [0.217, 0.334] |
| | Will | 0.198 [0.144, 0.253] | 0.209 [0.155, 0.263] | 0.166 [0.117, 0.215] | 0.485 [0.417, 0.552] | 0.373 [0.307, 0.439] |
| Qwen3-TTS-Flash | Cherry | 0.233 [0.175, 0.291] | 0.212 [0.158, 0.266] | 0.141 [0.096, 0.186] | 0.267 [0.207, 0.326] | 0.253 [0.195, 0.310] |
| | Ethan | 0.217 [0.162, 0.272] | 0.229 [0.173, 0.286] | 0.283 [0.222, 0.345] | 0.259 [0.202, 0.316] | 0.207 [0.154, 0.259] |

