# OpenReview forum: "Audio Turing Test: Benchmarking the Human-likeness of Large Language Model-based Text-to-Speech Systems in Chinese"
_ICLR.cc/2026/Conference — Submitted to ICLR 2026_

### Official Review · Reviewer_RQm4 · 2025-10-31

**Soundness:** 3
**Presentation:** 3
**Contribution:** 3
**Rating:** 4
**Confidence:** 3

**Summary:**

This paper proposes the Audio Turing Test (ATT), a novel evaluation framework for assessing the human-likeness of LLM-based Chinese Text-to-Speech (TTS) systems. ATT integrates a multi-dimensional corpus (ATT-Corpus) covering five key linguistic dimensions (e.g., Chinese-English code-switching, polyphonic characters), a Turing Test-inspired human evaluation protocol, and an automatic evaluation tool (Auto-ATT) fine-tuned on Qwen2.5-Omni-7B. Unlike traditional Mean Opinion Score (MOS) methods, ATT uses ternary judgments ([Human], [Unclear], [Machine]) to reduce bias and improve discriminative power. Experiments with 857 native Chinese listeners and 5 state-of-the-art TTS models show that ATT effectively distinguishes model performance, with top-performing Seed-TTS achieving a Human-likeness Score (HLS) of only 0.4—revealing gaps between synthetic and human speech. Auto-ATT demonstrates strong alignment with human judgments and outperforms conventional MOS predictors on trap items.

**Strengths:**

Targeted Solution to Critical Gaps: Addresses MOS’s limitations (subjectivity, low interpretability) and the lack of multi-dimensional, Chinese-specific TTS evaluation datasets, filling a key niche in LLM-driven TTS assessment.
Comprehensive Framework Design: Combines a well-constructed corpus (semi-automated generation + expert validation), rigorous human evaluation (trap items, consistency checks), and an efficient automatic tool, enabling both qualitative and quantitative analysis.
Robust Experimental Validation: Large-scale human evaluations (857 participants) and statistical tests (GLMM) confirm ATT’s reliability, while Auto-ATT’s superior performance over UTMOSv2 and DNSMOS Pro highlights its practical value for rapid model iteration.
Actionable Insights: Identifies specific weaknesses of current TTS systems (e.g., prosodic unnaturalness, flat emotional expression) and provides fine-grained comparisons across models, voices, and linguistic dimensions.

**Weaknesses:**

Language and Scenario Limitation: The framework is exclusively designed for Chinese, limiting generalizability to other languages with distinct linguistic features (e.g., tonal vs. non-tonal languages).
Narrow Trap Item Diversity: While trap items monitor attention, the paper only mentions "deliberately flawed synthetic clips" and "genuine human recordings"—more diverse trap types (e.g., edge-case linguistic structures) could strengthen robustness.
Auto-ATT Training Data Opacity: The paper references "additional private evaluation data" for Auto-ATT training without detailing its size, distribution, or how it complements public ATT-Corpus, raising questions about reproducibility.
Lack of Longitudinal or Real-World Testing: Evaluations focus on controlled audio clips; performance in real-world scenarios (e.g., background noise, dialogue context) is not explored, limiting insights into practical applicability.

**Questions:**

Given ATT’s Chinese-specific design, what key adaptations would be required to extend the framework to non-tonal languages (e.g., English) or languages with unique prosodic features (e.g., Japanese)?
How does Auto-ATT’s performance degrade when evaluating TTS models not included in its training data (e.g., newly developed models with novel architectures), and what strategies could mitigate this?

---

> ### Author Response · Authors · 2025-11-25
>
> We thank the reviewer for the detailed and constructive comments. We respond to each weakness and question below, and have incorporated the suggested clarifications and additional analyses into the revised version.
>
> > Weakness 1
> > Language and Scenario Limitation: The framework is exclusively designed for Chinese, limiting generalizability to other languages with distinct linguistic features (e.g., tonal vs. non-tonal languages).
> > Question 1
> > Given ATT’s Chinese-specific design, what key adaptations would be required to extend the framework to non-tonal languages (e.g., English) or languages with unique prosodic features (e.g., Japanese)?
>
> We agree that the current instantiation of ATT is Chinese-specific and we appreciate the opportunity to clarify which parts of the framework are language-agnostic and which are language-dependent.
>
> ATT is a general protocol for evaluating the human-likeness of speech synthesis. The Turing-test-style ternary decision ([Human]/[Unclear]/[Machine]), the use of native speakers as annotators, the trap-item based quality control, and the Human-likeness Score (HLS) definition do not rely on any property of Chinese in particular. The same protocol can be applied to any target language as long as the annotators are native speakers of that language.
>
> ATT implicitly evaluates four broad dimensions of human-likeness that are shared across languages: (i) segmental accuracy (are phones and words pronounced correctly), (ii) prosodic timing and intonation (pauses, rhythm, phrasing), (iii) paralinguistic and anthropomorphic behavior (emotion, expressivity, breath, laughter, etc.), and (iv) basic audio quality and clarity. These aspects are relevant whether the language is tonal (Chinese), stress-timed (English), or mora-timed (Japanese). The Chinese-specific corpus simply instantiates these generic dimensions with phenomena that are important for Chinese TTS.
>
> To extend ATT to other languages, we would keep the protocol and scoring unchanged, but redesign the capability subsets and text prompts in collaboration with linguists and experienced native speakers of the target language. For example:
>
> - For **English** (non-tonal), one would focus on phenomena such as reading of numbers, dates, and abbreviations; lexical stress patterns and stress shifts (e.g., “CONtract” vs. “conTRACT”); homographs and heteronyms; and code-switching with other languages (e.g., English–Spanish or English–Japanese). Classical or literary texts (e.g., Shakespearean sonnets) can play a role analogous to classical Chinese poetry in probing rhythm, meter, and phrasing.
> - For **Japanese** (mora-timed with pitch accent), one would introduce materials that stress mora timing, long vs. short vowel and consonant contrasts, pitch-accent patterns, and the pronunciation of kanji vs. kana and loanwords. Code-switching with English, and expressive reading of genres such as anime-style dialogue or haiku, would probe prosody and paralinguistic expressivity in a similar fashion.
>
> In all cases, the adaptation effort lies in constructing language-appropriate capability subsets and corpora with experts for that languages; the ATT protocol, trap-item design, and HLS definition remain unchanged and novel.
>
> > Weakness 2
> > Narrow Trap Item Diversity: While trap items monitor attention, the paper only mentions "deliberately flawed synthetic clips" and "genuine human recordings"—more diverse trap types (e.g., edge-case linguistic structures) could strengthen robustness.
>
> In ATT, trap items are intentionally designed with a narrow scope: their primary purpose is **attention and quality control for annotators**. We therefore use two easily recognizable categories: (i) clearly flawed synthetic clips (with obvious artifacts) and (ii) clean human recordings. This choice ensures that the “correct” answer for traps is unambiguous. When a participant fails such traps, it is highly likely due to inattention or random clicking, so excluding their batch improves data quality with minimal risk of discarding good judgments.
>
> Edge-case linguistic or prosodic structures trap clips are inherently more ambiguous: even expert listeners may reasonably disagree, especially when the synthetic speech is high quality but not perfect. Using these as traps would blur the distinction between “careless annotator” and “genuinely hard example,” and could lead to incorrectly filtering out high-quality labels.
>
> We agree that exploring richer trap designs that remain low-ambiguity could further strengthen robustness, while we do not expect such extensions to change the main conclusions of this paper.

---

> > ### Author Response · Authors · 2025-11-25
> >
> > > Weakness 3
> > > Auto-ATT Training Data Opacity: The paper references "additional private evaluation data" for Auto-ATT training without detailing its size, distribution, or how it complements public ATT-Corpus, raising questions about reproducibility.
> >
> > Thank you for pointing out that our description of the additional training data for Auto-ATT was not sufficiently detailed.
> >
> > We clarify the role of this data: the additional “private evaluation data” is synthesized based on ATT-Corpus with internal TTS systems and is only used to train Auto-ATT. All HLS scores and all ATT benchmark conclusions are computed solely from the human annotations on the ATT-Corpus.
> >
> > We understand the reviewer's concern for reproducibility. Upon acceptance, we plan to release all ATT-Corpus texts and the Auto-ATT model weights. For audo clips and human annotations, some evaluated audio clips are synthesized via commercial or ToS-restricted APIs; for those cases, we are carefully checking redistribution permissions. Where release is permitted, we will provide audio clips with ratings; where redistribution is not permitted, we will still release the texts and human ratings so that the community can re-synthesize the audio clips. Therefore, the results in the paper can be fully reproduced by the community.
> >
> > > Weakness 4
> > > Lack of Longitudinal or Real-World Testing: Evaluations focus on controlled audio clips; performance in real-world scenarios (e.g., background noise, dialogue context) is not explored, limiting insights into practical applicability.
> >
> > We thank the reviewer for raising the concern about real-world and longitudinal evaluation.
> >
> > First, we would like to clarify that the capability dimensions and Corpus in ATT are explicitly designed around real TTS product scenarios rather than artificial toy examples. The five capability subsets (Special Characters and Numerals, Chinese–English code-switching, Paralinguistic
> > Features and Emotions, Classical Chinese Poetry/Prose, Polyphonic Characters) are all motivated by concrete failure modes observed in deployed Chinese TTS systems, and the ATT-Corpus are adapted from or closely mimic realistic application contexts. In this sense, the evaluation already targets **practical, real-world application challenges** faced by Chinese TTS systems, and the resulting benchmark is directly informative for improving real applications.
> >
> > At the same time, we argue that the reviewer’s notion of “longitudinal or real-world testing” **goes beyond the TTS module itself** and leans toward end-to-end speech experience in noisy, interactive, or long-term settings. Our paper focus on evaluating one core fundamental capability of TTS: the human-likeness, and the above settings are conceptually distinct from the core TTS human-likeness and are out of the score of TTS system evaluation.

---

> ### Author Response · Authors · 2025-11-25
>
> > Question 2
> > How does Auto-ATT’s performance degrade when evaluating TTS models not included in its training data (e.g., newly developed models with novel architectures), and what strategies could mitigate this?
>
> We include two kinds of experiments designed precisely to probe this.
>
> (i) At the **voice-style level**, we hold out one voice per TTS family during Auto-ATT training and evaluate only on these unseen voices. Table 3 shows that Auto-ATT maintains very strong alignment with human judgments across both in-distribution and held-out capability dimensions: SRCC is around 0.9 and PLCC around 0.89–0.93, consistently outperforming the un-finetuned Qwen2.5-Omni baseline. This indicates that Auto-ATT does not simply memorize specific voices seen during training.
>
> (ii) At the **tts-system level**, Appendix E.2 in the revised paper version evaluates Auto-ATT on two completely unseen TTS model families (ElevenLabs Eleven v3 and Qwen3-TTS-Flash) that are not used in training. As shown in the below table, on the two capability dimensions that are entirely held out from training (Classical Chinese Poetry/Prose and Polyphonic Characters), Auto-ATT still achieves SRCC / PLCC scores of approximately 0.714 / 0.886 and 0.771 / 0.790, respectively. These correlations are strong, suggesting that Auto-ATT generalizes reasonably well even to TTS systems and voices it has never encountered before.
>
> | OOD Capability Dimension        | Auto-ATT SRCC | Auto-ATT PLCC | Qwen2.5 Omni SRCC | Qwen2.5 Omni PLCC |
> |--------------------------------|--------------:|--------------:|------------------:|------------------:|
> | Classical Chinese Poetry/Prose | 0.714         | 0.886         | -0.829            | -0.720            |
> | Polyphonic Characters          | 0.771         | 0.790         | -0.657            | -0.491            |
>
>
> Overall, these results indicate that Auto-ATT does exhibit some performance degradation under cross-system distribution shift, as expected, but the degradation is moderate rather than catastrophic: correlations remain high enough for Auto-ATT to serve as a useful proxy for human judgments on new TTS systems.
>
> To further mitigate potential degradation on future, unseen models, we see several promising directions as future works:
>
> 1. Expanding training diversity. Continually incorporating more TTS families with diverse architectures and speaking styles into the training pool would reduce the distribution gap between known and future systems, making Auto-ATT more generalizable.
>
> 2. Light-weight adaptation to new systems. For a newly deployed TTS model, collecting a small number of human-labeled examples and using them for light fine-tuning or calibration of Auto-ATT (e.g., a short LoRA update) could quickly adapt the evaluator to that system.
>
> We hope these clarifications address the reviewer’s concerns, and we would be grateful if the reviewer could consider raising the rating scores in light of the above points.

---

### Official Review · Reviewer_HQSS · 2025-11-01

**Soundness:** 2
**Presentation:** 3
**Contribution:** 3
**Rating:** 6
**Confidence:** 3

**Summary:**

This paper proposes Audio Turing Test (ATT), a human-likeness evaluation framework for Chinese LLM-TTS that pairs (i) a multi-dimensional corpus (ATT-Corpus) spanning numerals/special characters, code-switching, paralinguistics & emotions, classical prose/poetry, and polyphonic characters, with (ii) a ternary, Turing-style human protocol that labels each clip as Human / Unclear / Machine and derives a Human-Likeness Score (HLS) (1.0, 0.5, 0.0). The authors also fine-tune Qwen2.5-Omni-7B with human judgments to create Auto-ATT, a model-as-a-judge that predicts HLS and reportedly aligns strongly with human ratings. Benchmarks across five model families show clear separation and notably low absolute human-likeness (best model ≈0.4), in contrast with high MOS reported elsewhere.

**Strengths:**

* Multidimensional corpus targets common Chinese difficulty factors (polyphony, poetry syntax, code-switching).

* Ternary human protocol + rationales is a simple but meaningful shift away from MOS.

* The implementation of trap items as a good quality control.

* Auto-ATT is a useful direction; training a speech-judge model is under-explored, and the demonstrated correlation to humans is promising.

* The benchmark highlights meaningful gaps between SOTA models and human speech.

**Weaknesses:**

1. (Interpretability of Human-Likeness) HLS collapses three distinct cases (Human mistaken as Machine, Machine mistaken as Human, and Unclear) into a single linear score. Without reporting how often each category is chosen, it is unclear whether high HLS reflects genuine human-likeness or annotator uncertainty. Excessive “Unclear” selections may artificially inflate scores.

2. (Filtering Bias From Manual Spot Checks) The authors state that samples failing “synthesis success” or “synthesis consistency” are verified. Eliminating weak samples before evaluation can bias results toward best-case outputs. The paper does not clarify what was done with samples that failed this spot check.

3. (Sampling Policy Ambiguity) It is unclear whether annotators see one sample from each system or a random subset. If some participants repeatedly select “Unclear,” this may distort HLS. Details on randomization, system coverage, and balancing are not reported.

4. (Annotator and Expert Clarity) The paper inconsistently reports annotator counts (437 vs 857). Expert selection criteria are unclear, and post-hoc alignment of participant justifications to labels is subjective. The number of experts per sample and conflict resolution process are not specified.

5. Missing Citations -
* Praveen S V, Sherry Thomas, Sai Teja M S, Suvrat Bhooshan, Mitesh M. Khapra "The State Of TTS: A Case Study with Human Fooling Rates." Proc. Interspeech 2025
* Nguyen, Binh, and Thai Le. "TURING’S ECHO: Investigating Linguistic Sensitivity of Deepfake Voice Detection via Gamification." Proc. Interspeech 2025

6. (No Comparison With Established Listening Tests) The benchmark is not compared against MUSHRA or CMOS for ranking fidelity. While ATT may separate systems well, this has not been demonstrated relative to standard perceptual tests.

**Questions:**

1. Regarding manual spot checks: “We examine synthesis success and consistency.” What is the removal policy for failed samples? How many samples were discarded per system? Could this bias the evaluation toward cherry-picked successes?

2. How are samples assigned to annotators? Does each participant hear all systems so that within-subject comparison is possible?

3. How often do annotators choose Human / Machine / Unclear? Does widespread “Unclear” bias the HLS scale?

4. How were expert reviewers selected, and how many participated? In cases of disagreement regarding the presence or absence of an artifact, what adjudication procedure was applied?

5. The annotator count is inconsistently reported as 437 and 857. Which value is correct?

---

> ### Author Response · Authors · 2025-11-25
>
> We thank the reviewer for the careful reading and constructive suggestions. We have revised the paper to clarify the evaluation protocol, annotator and expert roles, the use of the [Unclear] label in HLS, and the relationship to MOS and concurrent work. Below we respond to each point in turn.
>
> > Weakness 1:
> > (Interpretability of Human-Likeness) HLS collapses three distinct cases (Human mistaken as Machine, Machine mistaken as Human, and Unclear) into a single linear score. Without reporting how often each category is chosen, it is unclear whether high HLS reflects genuine human-likeness or annotator uncertainty. Excessive “Unclear” selections may artificially inflate scores.
>
> > Question 3:
> > How often do annotators choose Human / Machine / Unclear? Does widespread “Unclear” bias the HLS scale?
>
> We appreciate this concern and have clarified both the design motivation for HLS and the empirical usage of [Unclear].
>
> (1) HLS design. We adopt a ternary label set {Human, Unclear, Machine} and map them to {1, 0.5, 0} to reflect the intended semantics of the Audio Turing Test: the annotator must decide whether the clip is human, synthetic, or genuinely ambiguous. The [Unclear] label is meant for borderline cases where annotators perceive strong human-likeness but remain uncertain, not as a default “don’t know” option.
>
> (2) New statistics on [Unclear] usage. To directly address the reviewer’s concern, we have added a detailed analysis of annotator-level [Unclear] rates in Appendix B.5. Empirically, [Unclear] is rarely used and is highly concentrated in a small subset of annotators:
>
> - Among the 857 annotators, 565 (65.93%) never selected [Unclear] at all.
> - 655 annotators (76.43%) have an unclear rate ≤ 5%, and 728 (84.95%) have an unclear rate ≤ 10%.
> - Only 93 annotators (10.85%) fall in the 10%–30% range.
> - The median unclear rate is 0.00%, with a mean of 4.62%.
>
> These statistics show that [Unclear] is not a dominant choice: most annotators provide decisive Human/Machine labels for nearly all clips. Thus, high HLS values primarily reflect genuine human-likeness rather than widespread reliance on [Unclear]. We now explicitly point readers to Appendix B.5 from Sec. 3.4 to make this clearer.
>
> > Weakness 2:
> > (Filtering Bias From Manual Spot Checks) The authors state that samples failing “synthesis success” or “synthesis consistency” are verified. Eliminating weak samples before evaluation can bias results toward best-case outputs. The paper does not clarify what was done with samples that failed this spot check.
>
> > Question 1:
> > Regarding manual spot checks: “We examine synthesis success and consistency.” What is the removal policy for failed samples? How many samples were discarded per system? Could this bias the evaluation toward cherry-picked successes?
>
> The intention of the manual spot checks is to catch _engineering-level_ failures, not to filter out low-quality but valid TTS outputs.
>
> We have revised Sec. 3.2 and Appendix A.2 to state this more explicitly:
>
> - We randomly sample 25% of synthesized clips _per system_ for expert spot checks.
> - This validation stage is “primarily intended to confirm that no widespread synthesis failures occur due to engineering issues or other extraneous factors” (e.g., network error).
> - Crucially, as now stated in Sec. 3.2: “Occasional synthesis failures at the level of a single audio clip are recorded but are not discarded at this stage.” In other words, **individual weak or failed clips are kept in the evaluation pool**, because they are part of the TTS system’s actual behavior.
>
> If spot checks had revealed a systematic engineering bug, we would have fixed the pipeline and regenerated that batch, rather than selectively discarding particular clips.
>
> Therefore, the evaluation is not biased by cherry-picking successes; all valid outputs, including occasional failures, are retained for human evaluation.

---

> ### Author Response · Authors · 2025-11-25
>
> > Weakness 3:
> > (Sampling Policy Ambiguity) It is unclear whether annotators see one sample from each system or a random subset. If some participants repeatedly select “Unclear,” this may distort HLS. Details on randomization, system coverage, and balancing are not reported.
>
> > Question 2:
> > How are samples assigned to annotators? Does each participant hear all systems so that within-subject comparison is possible?
>
> Thank you for pointing out the need to clarify the sampling policy. We have rewritten Sec. 3.3 to more clearly describe sampling and assignment.
>
> - Each participant receives 10 clips per evaluation batch: 7 evaluation clips and 3 trap items.
> - The 7 evaluation clips are “randomly sampled without replacement from a pool containing the synthesized audio clips for evaluation” (Sec. 3.3). This pool contains clips from all TTS systems and capability dimensions.
> - The 3 trap items consist of one deliberately flawed synthetic clip and two human recordings, also randomly assigned.
>
> Thus, each participant hears a _random subset_ of systems and capability settings; there is no guarantee that every participant hears all systems.
>
> Regarding within-subject comparisons: we intentionally _do not_ enforce a within-subject design where each listener rates all systems. This is for three reasons:
>
> 1. Avoid anchoring: if listeners continuously hear the same systems, their internal reference may drift and amplify anchoring effects, especially when systems are close to human quality.
> 2. Broader coverage under fixed budget: given the large number of voices x capabilities, a within-subject design would require each listener to rate very many clips, dramatically increasing fatigue or forcing us to subsample the space.
> 3. Statistical modeling via mixed effects: instead of relying on within-subject panels, we model participants as random effects in a generalized linear mixed model (GLMM) (Sec. 4.1.1). This allows us to control for individual rating tendencies and still obtain stable model-level estimates.
>
> As discussed above in response to Weakness 1 and in Appendix B.5, the [Unclear] option is infrequently used and concentrated in a small subset of annotators. Together with the GLMM analysis, this helps ensure that occasional heavy [Unclear] usage by a few participants does not distort system-level HLS.

---

> ### Author Response · Authors · 2025-11-25
>
> > Weakness 4:
> > (Annotator and Expert Clarity) The paper inconsistently reports annotator counts (437 vs 857). Expert selection criteria are unclear, and post-hoc alignment of participant justifications to labels is subjective. The number of experts per sample and conflict resolution process are not specified.
>
> > Question 4:
> > How were expert reviewers selected, and how many participated? In cases of disagreement regarding the presence or absence of an artifact, what adjudication procedure was applied?
> > Question 5:
> > The annotator count is inconsistently reported as 437 and 857. Which value is correct?
>
> 1. Annotators
>
> Annotator counts (437 vs. 857). Both numbers are correct but refer to _different_ studies:
>
> - **857 annotators** (Sec. 4.1) participated in the main ATT human evaluation. They produced one ternary judgment (Human / Unclear / Machine) plus a text justification for each assigned clip.
> - **437 annotators** (Sec. 3.5) are a _separate_ group used only for training Auto-ATT. They labeled additional training clips (including speech synthesized by internal TTS systems) with the same protocol.
>
> We have revised Sec. 3.5 and Sec. 4.1 to emphasize that these are two distinct annotator pools.
>
> 2. Experts
>
> We have defined experts explicitly in Sec. 3.1 and footnote 1: “Experts refer to individuals holding a master’s degree in linguistics or a related field.” In total, for the tasks discussed in Sec. 3.1–3.3, **two experts** are involved:
>
> - In Sec. 3.1, they review and cross-check the text corpus.
> - In Sec. 3.2, they perform the 25% audio spot checks for synthesis success and consistency.
> - In Sec. 3.3, they conduct the post-hoc consistency review of participants’ justifications.
>
> For both corpus validation and justification review, each sample is inspected by both experts. We adopt a conservative adjudication rule: a sample (or response batch) is accepted only if it passes both experts’ checks; if either expert flags an issue (e.g., obvious artifact in the corpus stage; justification not evidence-based or clearly unrelated in the response stage), that sample or response batch is excluded. We have added this conflict-resolution rule to Sec. 3.3.
>
> 3. Post-hoc alignment of justifications.
>
> Our procedure is limited to quality check of annotator behavior, not to changing labels:
>
> - Experts examine free-text justifications for the 7 non-trap synthetic clips in each batch, checking whether the explanation is concrete and grounded in perceptual evidence.
> - Responses whose justifications are vague, copied, or clearly unrelated (e.g., “just a guess”) are treated as inattentive and are removed at the batch level.
> - We do **not** modify the label itself based on expert judgment at this stage.
> - This process is more objective than subjective, as it focuses on the basic checking of concrete reasoning.
>
> > Weakness 5:
> > Missing Citations -
> > - Praveen S V, Sherry Thomas, Sai Teja M S, Suvrat Bhooshan, Mitesh M. Khapra "The State Of TTS: A Case Study with Human Fooling Rates." Proc. Interspeech 2025
> > - Nguyen, Binh, and Thai Le. "TURING’S ECHO: Investigating Linguistic Sensitivity of Deepfake Voice Detection via Gamification." Proc. Interspeech 2025
>
> We thank the reviewer for pointing us to these relevant concurrent works. We have added both citations and briefly discussed their relation to ATT in the Related Work.
>
> > Weakness 6:
> > (No Comparison With Established Listening Tests) The benchmark is not compared against MUSHRA or CMOS for ranking fidelity. While ATT may separate systems well, this has not been demonstrated relative to standard perceptual tests.
>
> We do not currently include MUSHRA or CMOS conditions since MUSHRA/CMOS designs are best suited to compare small number of systems evaluated jointly (CMOS usually compare 1 system with human), often with explicit human references and often do not produce a ranking among the systems, whereas ATT targets many TTS systems, voices and capability dimensions, and avoids explicit human references to reduce anchoring.
>
> In the current work, we have focused on MOS as the most widely used subjective metric, and we now make that comparison more explicit.
>
> In Sec. 4.1.1 and Appendix C, we report a MOS-based listening test on the same systems, analyzed via GLMM (Table 7). As highlighted in the revised text, we find:
>
> - Strong consistency between HLS and MOS in ranking systems: higher HLS models also achieve higher MOS.
> - However, HLS exhibits a substantially higher signal-to-noise ratio (SNR) than MOS (10.53 vs. 5.79), indicating greater separability across models and, by implication, lower annotator burden for achieving the same statistical power.
>
> We hope these clarifications address the reviewer’s concerns, and we would be grateful if the reviewer could consider raising the rating scores in light of the above points.

---

### Official Review · Reviewer_C953 · 2025-11-02

**Soundness:** 3
**Presentation:** 3
**Contribution:** 3
**Rating:** 6
**Confidence:** 3

**Summary:**

The authors propose the AudioTuringTest Benchmark, a new evaluation framework for Chinese TTS models. The paper attempts to address the reproducibility and saturation issues of MOS-like metrics, the de-facto evaluation protocol for TTS models. To do this, the ATT benchmark attempts to simplify the rating criteria into a Turing Test-like metric, whether or not a speech sample is from a human. ATT is created with human judgements of synthetic and real data. The authors also create Auto-ATT a model-as-a-judge version of ATT. Results suggest that ATT is able to capture key differences between TTS systems along several axes and Auto-ATT correlates well with human judgement.

**Strengths:**

- ATT attempts to address the critical limitations of MOS / pseudo-MOS by disentagling speech characteristics at the data level and simplifying the evaluation scheme
- ATT evaluates along several axes, such as numerals, code-switching, paralinguistics, and poetry.
- ATT can clearly distinguish the strengths and weakness of different model along each axes, allowing fine-grained insights of TTS performance
- Auto-ATT is a novel model-as-judge that can be used to automate the application of ATT at scale

**Weaknesses:**

- The ATT corpus is developed using the TTS models the authors intend on evaluating. It is unclear how it and AutoATT generalize to unseen systems, which does not address the claimed robustness issue of pseudo-MOS.
- ATT cannot distinguish speaker-level characteristics, which makes evaluation using speaker similiarity MOS or neural embeddings still required

**Questions:**

- Will the annotator ratings be released with the dataset?

---

> ### Author Response · Authors · 2025-11-25
>
> We thank the reviewer for the careful reading and for raising these important points. We respond below.
>
> > Weakness 1. “The ATT corpus is developed using the TTS models being evaluated. It is unclear how it and Auto-ATT generalize to unseen systems, which does not address the claimed robustness issue of pseudo-MOS.”
>
> We appreciate this concern and would like to clarify our setting.
>
> (1) ATT-Corpus and evaluated audio clips are two different components. The ATT-Corpus is a fixed set of texts that targets five key Chinese TTS capability dimensions, as described in Section 3.1. The evaluated audio clips are synthesized by different TTS systems from this corpus only at evaluation time (Section 3.2).
>
> (2) The ATT-Corpus construction does not use any evaluated TTS system. As detailed in Section 3.1 and Appendix A.1, corpus texts are produced via a semi-automatic pipeline: GPT-4o generates dimension-specific drafts, DeepSeek-R1 performs colloquial adaptation, and four linguistics experts conduct standardized revision and cross-checking. No TTS model outputs are involved in generating the corpus.
>
> (3) The ATT protocol is model-agnostic and can be applied to unseen systems. Under ATT, annotators judge each audio clip with a ternary authenticity label [Human / Unclear / Machine], accompanied by trap items and justification-based validation (Section 3.3). Because the task asks directly whether a clip sounds human, the protocol does not depend on any specific model family, reference speaker, or prior system comparison. This design helps avoid the scale drift and anchoring effects commonly observed in MOS-style evaluations.
>
> (4) Auto-ATT is evaluated under held-out generalization settings. Auto-ATT is trained on a corpus-level split (three training dimensions vs. two held-out dimensions) and a strict voice-style-level held-out split within each of the five model families (Section 3.5 and Appendix G). The resulting predictions show strong alignment with human judgments on both in-distribution and out-of-distribution dimensions (Table 3), indicating robust generalization to unseen voice styles and capability subsets.
>
> (5) Beyond the originally submitted version, we performe an additional and stricter *unseen TTS system* generalization test to directly address robustness to newly emerging TTS models. Concretely, on the two held-out OOD capability dimensions, we introduce two recently released TTS systems that are entirely absent from Auto-ATT training: ElevenLabs Eleven v3 and Qwen3-TTS-Flash, and synthesize evaluation clips from the held-out ATT-Corpus test set. As shown in the below table, Auto-ATT remains strongly aligned with human judgments on both OOD dimensions, yielding high positive rank and linear correlations. These results provide additional evidence that ATT-Corpus and Auto-ATT generalize well to *newly appearing* TTS systems without any system-specific tuning.
>
> | OOD Capability Dimension        | Auto-ATT SRCC | Auto-ATT PLCC | Qwen2.5 Omni SRCC | Qwen2.5 Omni PLCC |
> |--------------------------------|--------------:|--------------:|------------------:|------------------:|
> | Classical Chinese Poetry/Prose | 0.714         | 0.886         | -0.829            | -0.720            |
> | Polyphonic Characters          | 0.771         | 0.790         | -0.657            | -0.491            |
>
>
> (6) ATT provides higher separability than MOS in our experiments. Beyond protocol-level robustness, we observe that the Human-likeness Score (HLS) yields a substantially higher signal-to-noise ratio than MOS (10.53 vs. 5.79), implying clearer discrimination across systems (Section 4.1.1).
>
> > Weakness 2. “ATT cannot distinguish speaker-level characteristics, which makes evaluation using speaker similarity MOS or neural embeddings still required.”
>
> We agree that speaker-level evaluation is valuable, and we respectfully note that ATT is designed for a different goal. ATT measures general human-likeness—whether synthesized speech resembles real human speech—rather than similarity to a specific target speaker. Speaker similarity metrics (e.g., SIM or neural-embedding-based scores) quantify speaker identity fidelity, while ATT focuses on human-likeness factors such as prosodic naturalness, contextual expressivity, and audio artifacts. These two evaluations address different aspects of TTS quality and can be used in a complementary manner.

---

> > ### Author Response · Authors · 2025-11-25
> >
> > > Question. “Will the annotator ratings be released with the dataset?”
> >
> > Yes. We plan to release annotator ratings for the white-box portion of the dataset together with the corresponding texts and trap items. Some evaluated audio clips are synthesized via commercial or ToS-restricted APIs; for those cases, we are carefully checking redistribution permissions. Where release is permitted, we will provide audio clips with ratings; where redistribution is not permitted, we will still release the texts and human ratings so that the community can reproduce the evaluation protocol and compare systems under the same Human-likeness Score setup.
> >
> > We hope these clarifications address the reviewer’s concerns, and we would be grateful if the reviewer could consider raising the rating scores in light of the above points.

---

### Author Response · Authors · 2025-11-25
**General Response**

We have submitted a revised version of the paper, where all modifications are highlighted in blue.

The main additions and clarifications are as follows:

1. Related works.
   We expanded the Related Works section by adding recent references on the impact of task framing on subjective evaluation and on human-centered TTS evaluation. This situates our contribution more clearly within the existing literature.

2. Annotation protocol and quality control.
   - We clarify that 2 experts with a master’s degree in linguistics (or related fields) independently review the free-text justifications for consistency, and that labels failing this expert check are filtered out according to strict criteria.

3. Statistics on the use of the [Unclear] label.
   We added an analysis of how often annotators chose the [Unclear] option. Approximately two-thirds of annotators never used [Unclear], and overall usage is very low, which alleviates concerns that annotators might be overusing [Unclear] as an “escape” option.

4. Auto-ATT training / testing data split clarification.
   - We clearly distinguish the capability subsets used for training Auto-ATT from those reserved exclusively for evaluation as out-of-distribution (OOD) subsets.
   - We list, for each TTS family, the specific held-out voice style that is used only for testing, ensuring that there is no information leakage between training and evaluation for Auto-ATT.

5. New generalization experiments on unseen TTS systems.
   - We added experiments on two previously unseen TTS systems, ElevenLabs Eleven v3 and Qwen3-TTS-Flash, where we re-collected human annotations on ATT prompts and evaluated Auto-ATT on their synthesized audio.
   - On Out-of-distribution capability dimensions (Classical Chinese Poetry/Prose and Polyphonic Characters), Auto-ATT maintains high correlation with human judgments (e.g., SRCC ≈ 0.71–0.77, PLCC ≈ 0.79–0.89), whereas the original Qwen2.5-Omni exhibits negative correlations under the same setting. These results further support the robustness and generalization ability of Auto-ATT.

---

### Author Response · Authors · 2025-12-03

# Summary Response

We sincerely thank the AC and all three reviewers for their thorough reading and constructive feedback. We especially appreciate the AC's additional efforts under the special circumstances. We have comprehensively revised the paper and conducted additional experiments in response to the comments.

## 1. Background and Core Contributions

Current LLM-driven TTS systems are rapidly approaching human-level naturalness, yet traditional metrics such as MOS suffer from subjective drift, dimensional conflation, and ceiling effects, making it difficult to accurately characterize human-likeness—particularly in Chinese, where no multi-dimensional evaluation framework exists. To address these challenges, we propose the **Audio Turing Test (ATT)**, which encompasses five core contributions:

1. **Evaluation Protocol**: A Turing-test-style ternary annotation scheme (Human / Unclear / Machine) that is model-agnostic and reproducible, with trap-item mechanisms and expert quality control to ensure annotation reliability.
2. **Evaluation Dataset (ATT-Corpus)**: A corpus covering five key Chinese TTS capability dimensions (Special Characters and Numerals, Chinese–English Code-switching, Paralinguistic Features and Emotions, Classical Chinese Poetry/Prose, and Polyphonic Characters).
3. **Evaluation Metric (HLS)**: The Human-likeness Score, which achieves substantially higher signal-to-noise ratio than MOS (10.53 vs. 5.79).
4. **Automatic Scoring Model (Auto-ATT)**: A model fine-tuned from Qwen2.5-Omni-7B that enables rapid model-as-a-judge evaluation.
5. **Generalization Validation**: Demonstrated strong correlation with human judgments under held-out voice-style settings, out-of-distribution capability dimensions, and entirely unseen TTS systems.


## 2. Categorized Summary of Major Concerns and Responses

### (A) Framework Generalization, Language Scope, and Application Scenarios (Reviewer C953-W1 in part; Reviewer RQm4-W1, W4, Q1)

**Concerns**: Does ATT-Corpus depend on the evaluated models? Is the framework limited to Chinese? Is there a lack of real-world scenario testing?

**Response**: ATT-Corpus is a fixed text corpus generated by GPT-4o and DeepSeek-R1, then revised by four linguistics experts. It does not involve any TTS system outputs (Sec. 3.1). The protocol itself is a model-agnostic Turing-test-style judgment that does not depend on specific model families or reference speakers.

Regarding language extension, the protocol, HLS, and trap mechanisms are all language-agnostic. We have added examples for English (stress patterns, heteronyms) and Japanese (pitch accent, mora timing) to illustrate how ATT can be extended to other languages by retaining the protocol while adapting the corpus.

Regarding real-world scenarios, the five capability dimensions are derived from common failure modes observed in deployed Chinese TTS products, and the corpus texts closely mimic realistic application contexts. Evaluation of noisy environments or conversational interactions pertains to end-to-end speech experience and lies beyond the scope of TTS module human-likeness assessment.

### (B) Interpretability of HLS and the [Unclear] Label (Reviewer HQSS-W1, Q3)

**Concerns**: Could [Unclear] artificially inflate HLS? Does collapsing three labels into one dimension lose information?

**Response**: We have added detailed statistics on [Unclear] usage (Appendix B.5). Among 857 annotators, 65.93% never selected [Unclear], 76.43% had an unclear rate ≤5%, the median was 0.00%, and the mean was only 4.62%. These statistics demonstrate that high HLS values primarily reflect definitive Human judgments rather than annotator uncertainty. The [Unclear] label is semantically intended for cases where the speech is highly human-like yet some doubt remains—distinct from random or default selection.

---

> ### Author Response · Authors · 2025-12-03
>
> ### (C) Evaluation Protocol Details (Reviewer HQSS-W2, W3, W4, Q1, Q2, Q4, Q5; Reviewer RQm4-W2)
>
> **Concerns**: Do spot checks filter out weak samples? What is the sampling strategy? Are annotator/expert counts inconsistent? Are trap items too narrow?
>
> **Response**: Regarding spot checks (Sec. 3.2), the 25% random sampling is solely intended to detect systematic engineering failures; individual failed samples are retained rather than discarded, so there is no cherry-picking.
>
> Regarding sampling strategy (Sec. 3.3), each batch consists of 10 clips (7 evaluation + 3 trap), randomly sampled without replacement from the global pool. We intentionally do not require each participant to evaluate all systems, thereby avoiding anchoring effects; individual differences are controlled through GLMM modeling.
>
> Regarding personnel counts, the 857 annotators participated in the main human evaluation, while the 437 annotators form a separate group used only for Auto-ATT training (Sec. 3.5, 4.1). Four linguistics experts (holding master's degrees in linguistics or related fields) were responsible for corpus revision and cross-validation (Sec. 3.1), while two experts conducted audio spot checks (Sec. 3.2) and annotator justification consistency review (Sec. 3.3). A conservative adjudication rule is applied: if either expert flags an issue, the sample or response batch is excluded. Experts do not modify labels themselves; they only remove unreliable annotations.
>
> Regarding trap design, traps serve the purpose of attention and quality control rather than difficult-case testing. We intentionally use low-ambiguity designs (clearly flawed synthetic clips and clean human recordings) to avoid incorrectly filtering out attentive annotators.
>
> ### (D) Auto-ATT Training Data and Generalization (Reviewer C953-W1 in part; Reviewer RQm4-W3, Q2)
>
> **Concerns**: Can Auto-ATT generalize to unseen systems? The "private data" description is unclear. How does Auto-ATT performance degrade on new models?
>
> **Response**: The additional data consists only of audio synthesized by internal TTS systems on the training corpus, labeled by the 437 annotators, used to increase Auto-ATT training diversity. All HLS conclusions in the paper are based solely on ATT-Corpus and human annotations (Sec. 3.5, Appendix E.1).
>
> Auto-ATT employs a strict held-out setting: corpus-level (3 training + 2 OOD dimensions) and voice-style-level (one test voice per TTS family). We have added generalization experiments on entirely unseen TTS systems: on ElevenLabs Eleven v3 and Qwen3-TTS-Flash, Auto-ATT achieves SRCC of 0.714–0.771 and PLCC of 0.790–0.886 on OOD dimensions (Appendix E.2), indicating moderate rather than catastrophic degradation.
>
> We discuss future mitigation strategies, including continually incorporating more model families into training and performing lightweight LoRA adaptation with small-scale annotations for new systems.
>
> ### (E) Relationship to MOS and Related Work (Reviewer HQSS-W5, W6; Reviewer C953-W2)
>
> **Concerns**: Missing comparison with MOS/MUSHRA/CMOS; missing recent citations; relationship to speaker similarity.
>
> **Response**: Our MOS comparison experiment (Appendix C) shows that HLS and MOS rankings are consistent, but HLS achieves a higher signal-to-noise ratio (10.53 vs. 5.79), indicating better discriminability. MUSHRA and CMOS are better suited for local comparisons of a small number of systems with explicit human references, whereas ATT targets large-scale benchmarking across multiple families, voices, and capabilities, deliberately avoiding explicit human references to reduce anchoring effects.
>
> We have added citations to "The State of TTS" and "Turing's Echo" (both Interspeech 2025) and discussed their relation to ATT (Sec. 2).
>
> Regarding speaker similarity, ATT evaluates "whether speech sounds human" (focusing on prosody, emotion, and artifacts), while SIM evaluates "whether speech sounds like a specific person." These two objectives are distinct and complementary.
>
> ### (F) Data Availability (Reviewer C953-Q1)
>
> **Concern**: Will annotation data be released with the dataset?
>
> **Response**: Yes. We plan to release the white-box portion of ATT-Corpus texts, trap items, and corresponding human annotations. For audio clips synthesized via commercial or ToS-restricted APIs, we are verifying redistribution permissions case by case: where permitted, we will release audio clips with annotations; where not permitted, we will still release texts and annotations so the community can re-synthesize and reproduce the evaluation. Auto-ATT model weights and code will be released upon acceptance.

---

> ### Author Response · Authors · 2025-12-03
>
> ## 3. How the Revisions and Rebuttals Strengthen Core Contributions
>
> Through this revision, all five core contributions have been further strengthened.
>
> **Evaluation Protocol**: We have clarified the sampling strategy, trap design, and dual-expert quality control mechanism, substantially improving reproducibility and transparency while addressing concerns about subjective bias.
>
> **Evaluation Dataset**: We have clarified that corpus construction is entirely independent of evaluated models, added discussion on multi-language extension and real-world application scenarios, and committed to releasing the dataset and annotations.
>
> **Evaluation Metric**: The new [Unclear] usage statistics demonstrate that HLS semantics are clear and not dominated by uncertain labels; the systematic comparison with MOS confirms that HLS achieves higher signal-to-noise ratio and discriminability.
>
> **Automatic Scoring Model**: We have provided detailed training/testing split descriptions, improved training data transparency, and outlined open-source plans and future adaptation strategies.
>
> **Generalization Validation**: The new experiments on unseen TTS systems (Eleven v3, Qwen3-TTS-Flash) directly demonstrate the generalization capability of ATT-Corpus and Auto-ATT to emerging systems, substantively addressing the core concern about generalization.
>
> We believe these revisions adequately address the reviewers' concerns. ATT fills the gap in multi-dimensional test sets and standardized protocols for Chinese TTS human-likeness evaluation, providing an important foundation for advancing TTS toward more natural and human-like performance.
>
>
> ## 4. Summary of Revisions
>
> | Revision                                                                                                                                         | Location                    |
> | ------------------------------------------------------------------------------------------------------------------------------------------------ | --------------------------- |
> | Added recent related work citations (effect of task assumptions on subjective evaluation; human-centered TTS evaluation)                         | Sec. 2                      |
> | Added [Unclear] label usage statistics (approximately 2/3 of annotators never used it; overall usage rate is low)                                | Appendix B.5                |
> | Clarified expert definition (master's degree in linguistics) and conservative adjudication mechanism (exclusion if either expert flags an issue) | Sec. 3.1 footnote, Sec. 3.3 |
> | Clarified purpose of spot checks (detecting systematic engineering failures) and handling strategy (individual failed samples retained)          | Sec. 3.2, Appendix A.2      |
> | Rewrote sampling and assignment strategy (10 clips per batch sampled without replacement; individual differences controlled via GLMM)            | Sec. 3.3                    |
> | Distinguished two independent annotator groups (857 for main experiment vs. 437 for Auto-ATT training)                                           | Sec. 3.5, Sec. 4.1          |
> | Clarified Auto-ATT training/testing capability subset split (3 training + 2 OOD) and held-out voices for each TTS family                         | Sec. 3.5, Appendix E.1      |
> | Added generalization experiments on unseen TTS systems (Eleven v3, Qwen3-TTS-Flash; SRCC 0.714–0.771 on OOD dimensions)                          | Sec. 4.2.2, Appendix E.2    |
>
>
> We thank the AC and reviewers once again for their valuable feedback and hope that our revisions adequately address the concerns raised.

---

### Meta-Review · Area_Chair_sc78 · 2026-01-06

**Summary:**

This paper proposes a novel Turing-test–style subjective evaluation metric for Chinese text-to-speech (TTS) systems. The authors construct a crowdsourced dataset in which annotators perform such evaluations across multiple TTS systems and provide natural-language rationales for their judgments. A large multimodal language model (Qwen 2.5-Omni) is then fine-tuned to learn both the human ratings and the associated reasoning, with the goal of enabling automated analysis in the future. Experimental results suggest that the resulting Human-Likeness Score (HLS) correlates well with MOS, while exhibiting stronger discriminative power and higher signal-to-noise ratios.
However, several critical concerns raised by reviewers remain unresolved:

**1. Ecological validity of the evaluation content.**
Reviewers RQm4-W1, W4, and Q1 pointed out that the textual content used in the ATT corpus lacks important real-world context, such as spoken dialogue structure and background noise, which are essential characteristics of many practical TTS applications. This concern was not adequately addressed. The authors argue that incorporating such factors goes beyond the scope of TTS evaluation, but this claim is questionable. Spoken dialogue generation (e.g., for podcasts, conversational agents, and spoken dialogue systems) is a major application of current TTS systems, and evaluating performance in such contexts is therefore highly relevant.

**2. Limitations in speaker similarity assessment.**
Reviewer C953-W2 noted that ATT is not suitable for evaluating speaker similarity. The authors respond that ATT is intended to complement speaker similarity evaluation. However, this response highlights a more fundamental limitation of ATT: unlike MOS, it cannot be flexibly instructed to focus on specific perceptual dimensions (e.g., naturalness vs. similarity). This lack of controllability constrains its applicability as a general-purpose evaluation metric.

**3. Lack of comparison with other established subjective metrics.**
Reviewer HSQSS-W6 highlighted the absence of comparisons with widely adopted subjective evaluation protocols such as MUSHRA and CMOS. The authors did not provide corresponding experimental results and instead offered only limited qualitative analysis, leaving this concern largely unaddressed.

In addition, the Area Chair has the following concerns:

**1. Insufficient validation of ATT as a subjective metric.**
As a newly proposed subjective evaluation metric, ATT requires more comprehensive validation before it can be considered reliable. In particular, the consistency of HLS across evaluators is unclear, as is the distribution of HLS scores for real speech samples produced by different human speakers, which would serve as an important reference.

**2. Unclear source of improved discriminative power.**
A potential strength of ATT lies in its use of a carefully designed text corpus, which may reduce the risk of cherry-picked samples, which is a known issue in MOS-based studies. However, it remains unclear to what extent this factor, rather than the metric itself, contributes to the reported improvement in discriminative ability over MOS.

**3. Limited practical advantage over MOS and missing baselines.**
It is unclear how ATT better supports TTS evaluation and system development compared to MOS. The proposed metric produces essentially the same system ranking as MOS, while being less flexible due to its constrained instruction mechanism. Moreover, recent audio-LLM–based MOS prediction methods exist and would constitute more appropriate baselines than smaller models such as DNSMOS and UTMOS.

**Reviewer Concerns:**

W1, W4 and Q1 from Reviewer RQm4, W6 from Reviewer HQSS and W2 from Reviewer C953 remain unresolved, while the remaining issues have been adequately addressed.

**Reviewer Scores:**

I believe all three reviewers would maintain their original ratings, as several critical concerns remain unresolved in the rebuttal. Consequently, the reviews remain non-consensual.

---

### Decision · Program_Chairs · 2026-01-26

Reject